# THE LOGARITHM TRICK: ACHIEVE BETTER LONG TERM FORECAST VIA MEAN LOGARITHM SQUARE LOSS

## ABSTRACT

Weather forecasting and time series prediction can be modeled as autoregressive prediction tasks and optimized through a pretraining-finetuning paradigm. We discovered that simply incorporating an element-wise logarithmic operation following the standard square error loss, which we term MLSE, noticeably enhances long-term forecast performance in the fine-tuning phase. Remarkably, MLSE acts as a plug-and-play, zero-cost enhancement for autoregressive tasks. In this paper, we conduct a series of comprehensive experiments that support the effectiveness of MLSE. Furthermore, we present a phenomenological theory to dive into the feasibility and limitations of MLSE, by modeling the rate of error accumulation. Our findings propose a promising direction for understanding long-term prediction based on finite history.

## 1 INTRODUCTION

Autoregressive prediction tasks, such as series prediction, are commonly modeled as sequential tasks where each prediction relies on previous outputs. The goal of these models is to produce a coherent and consistent sequence of outputs, capturing the intricate dependencies within the data. Such models find wide applications in language modeling Radford et al. (2018), machine translation Vaswani et al. (2017), image generation Salimans et al. (2017), speech synthesis Oord et al. (2016) , and notably, weather forecasting.

This paper concentrates on time series prediction, such as weather forecasting. In such contexts, autoregressive machine learning (ML) models have demonstrated their efficacy by offering precise worldwide forecasts that outperform traditional methods. Most of these ML models Lam et al. (2022); Chen et al. (2023) adopt the pretrain-finetune paradigm. Initially, a model is pretrained on a two-time-stamp prediction task, and subsequently finetuned on a three or more time-stamp prediction task. As this paradigm optimizes the error per time stamp in a recurrent manner, these models are all prone to a well-known challenge called "error accumulation," which errors tend to accumulate as the forecast extends into the future, posing a significant challenge for long-term predictions.

Several efforts have been made to overcome the error accumulation problem. GraphCast Lam et al. (2022) rigorously extends the finetuning process to 12-stage predictions, while PanguWeather Bi et al. (2022) configures multiple models to directly model multi-length predictions, thereby bypassing the recurrent process. FengWu Chen et al. (2023), on the other hand, maintains a replay buffer to store the detached time state during prediction. However, all of these methods require significant additional resources to gain better control over long-term forecast errors.

In this paper, we present a simpler solution: by merely integrating an element-wise logarithmic operation after the standard Mean Square Error(MSE) loss - a strategy we term Mean Log Square Error (MLSE), we noticeably enhance the long-term forecast performance during the fine-tuning phase. As it only involves

a minor modification to the loss format, MLSE serves as a plug-and-play, zero-cost enhancement for autoregressive tasks.

This paper is organized as follows: First, we provide a brief overview of autoregressive modeling for sequential data and introduce the pretrain-finetune paradigm. Second, we present a series of experiments to validate the effects of the logarithmic operation. Third, we propose a phenomenological theory to analyze the rate of error accumulation, a concept we refer to as the error "amplifier". This theoretical perspective affords a deeper understanding of error behavior in autoregressive tasks. It inspires numerous advanced 'amplifier' tricks, including MLSE. Consequently, we conduct a series of ablation studies to identify the most efficient way. Finally, we analyze the applicability of our method and discuss the limitations of our theoretical framework.

## 2 AUTOGRESSIVE MODELING AND PRETRAIN-FINETUNE PARADIGM

The autoregressive procedure requires a parametrized model $f_\theta$ capable of accurately predicting the future state $\boldsymbol{X}_{t+n}^N$ at time $t + n$, given the historical data $\boldsymbol{X}_t^O$ at time $t$.

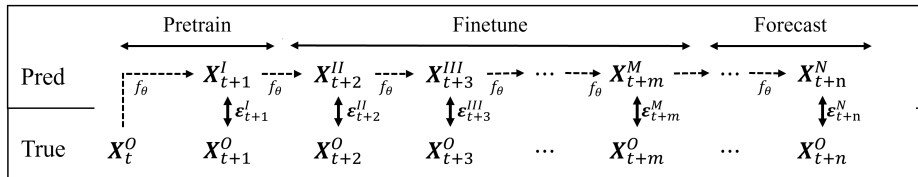

Figure 1: The diagram of pretrain-finetune paradigm for autoregressive procedure.

As illustrated in Fig.1, we can generate two sequences: the predicted states $\boldsymbol{X}_{t+1}^I, \boldsymbol{X}_{t+2}^{II}, \cdots, \boldsymbol{X}_{t+n}^N$ and the ground truth states $\boldsymbol{X}_t^O, \boldsymbol{X}_{t+2}^O, \cdots, \boldsymbol{X}_{t+n}^O$. Here, we use superscripts (in roman numerals) to represent the number of times $f_\theta$ is applied (denote the term 'order'), and subscripts to denote the actual timestamp. We use $m$ to indicate the timestamp used for model training and $n$ to denote the timestamp used for evaluating forecast performance. Each state $\boldsymbol{X}_{t+n}^N = [\boldsymbol{X}_{t+n,0}^N, \boldsymbol{X}_{t+n,1}^N, \cdots, \boldsymbol{X}_{t+n,K}^N]$ is a $K$-dimension vector.

The goal of series prediction tasks is to generate an accurate estimate of $\boldsymbol{X}_{t+n}^N$ for $n >> 1$. Typically, numerical weather forecasts consider $n \approx 20$, while time series predictions consider $n \approx 4$. The most straightforward approach is to train the model supervised from the near future ($m = 1$) to the further future ($m = n = 20$) directly, referred to the end-to-end-forecast paradigm for smaller systems Wu et al. (2022). However, for larger systems with large $n$, this method becomes impractical due to the large memory requirements and the slow training speed. Hence, researchers frequently adopt the pretrain-finetune paradigm. This approach initially involves training the model on the immediate next time stamp ($m = 1$) for a significant number of epochs, followed by finetuning the model on the subsequent timestamps ($m = 2$) over a few epochs. In other words, it optimizes both $\varepsilon_{t+1}^I$ and $\varepsilon_{t+2}^{II}$ during finetune.

In practice, there are various fine-tuning strategies to effectively minimize long-term future error. For example, one can extend finetuning to more time stamps, as demonstrated by Lam et al. (2022), who successfully extended the finetuning stage to $m = 12$ and observed improved performance with an increasing number of finetuning stages. Bi et al. (2022) configured multiple models to directly model multi-length predictions, thereby circumventing the recurrent process. On the other hand, Chen et al. (2023) maintained a replay buffer to store the detached time state during prediction, thus avoiding the need for the large gradient cache and achieving superior performance.

The error utilized for optimization and the error metric used for evaluation in all frameworks, denoted $[\varepsilon_{t+1}^I, \varepsilon_{t+2}^{II}, \cdots, \varepsilon_{t+m}^M, \cdots, \varepsilon_{t+n}^N]$. The Mean Squared Error (MSE), also known as the $L2$ loss, is defined

as $MSE_t^N = \epsilon_{t+n}^N = ||\varepsilon_{t+n}^N|| = \frac{1}{K}\sum_s [(\boldsymbol{X}_{t+N,s}^N - \boldsymbol{X}_{t+N,s}^O)^2]$. We found that it is possible to endow the model with the ability to perceive long-term future errors by optimizing in the near future. This leads to the development of the "Logarithm Trick", which simply integrates the logarithmic operation before computation of the average MSE loss, denoted by the MLSE loss.

$$MLSE_t^N = \frac{1}{K}\sum_s \ln[(\boldsymbol{X}_{t+N,s}^N - \boldsymbol{X}_{t+N,s}^O)^2] \tag{1}$$

## 3 EXPERIMENT

In this section, we will demonstrate how the series prediction task reaps benefits from the Mean Logarithmic Square Error (MLSE) within the pretraining-finetuning framework. The underlying reasons and the theoretical analysis explaining these benefits will be elaborated in Section 4. We will use two systems as examples: a numerical weather forecasting system and a time-series prediction task.

### 3.1 RESULT FOR WEATHER FORECASTING SYSTEM

Weather forecasting attempts to uncover the laws of latent physical evolution in large datasets. It divides the atmosphere of the globe into a grid and simulates the weather conditions at each grid point at different time steps. This implies that the transition between two time steps is determined by the entire global spatial information and can be formulated as an implicit neural network simulator such as FourCastNet Pathak et al. (2022). Weather forecast datasets, such as WeatherBench Rasp et al. (2020), are very large, making the training system complex and extensive. By examining the evolution of its physical fields, we can observe that changes in the state at each time stamp follow a comprehensive rule, exhibiting a certain degree of stability. As a result, the weather forecasting task is usually configured as a stamp-to-stamp task.

| Dataset | WeatherBench32x64-55k | | | | | WeatherBench32x64-300k | | | | | | WeatherBench64x128-300k | | | | | |
|---|---|---|---|---|---|---|---|---|---|---|---|---|---|---|---|---|---|
| Model | FourCastNet | | | | | FourCastNet | | | Lgnet | | | FourCastNet | | | Lgnet | | |
| T+m | T+1 | T+2 | | T+3 | | T+1 | T+2 | | T+1 | T+2 | | T+1 | T+2 | | T+1 | T+2 | |
| Loss | - | L2 | Ln | L2 | Ln | - | L2 | Ln | - | L2 | Ln | - | L2 | Ln | - | L2 | Ln |
| T850 n=4 | 1.17 | 1.09 | 1.01 | 1.08 | 1 | 1.10 | 1.03 | 0.97 | 0.92 | 0.89 | 0.85 | 0.83 | 0.77 | 0.76 | 0.75 | 0.70 | 0.70 |
| T850 n=12 | 1.75 | 1.6 | 1.52 | 1.57 | 1.49 | 1.68 | 1.56 | 1.45 | 1.46 | 1.38 | 1.34 | 1.42 | 1.30 | 1.27 | 1.34 | 1.19 | 1.16 |
| T850 n=20 | 2.52 | 2.34 | 2.28 | 2.29 | 2.23 | 2.48 | 2.39 | 2.23 | 2.20 | 2.11 | 2.05 | 2.18 | 2.02 | 1.99 | 2.07 | 1.90 | 1.85 |
| Z500 n=4 | 93 | 82 | 71 | 81.7 | 70 | 83 | 76 | 70 | 65 | 63 | 60 | 72 | 53 | 51 | 51 | 47 | 44 |
| Z500 n=12 | 256 | 225 | 213 | 220 | 209 | 218 | 203 | 196 | 192 | 183 | 177 | 180 | 162 | 159 | 160 | 145 | 140 |
| Z500 n=20 | 479 | 437 | 422 | 424 | 412 | 393 | 381 | 371 | 394 | 378 | 362 | 377 | 346 | 338 | 341 | 318 | 310 |

Table 1: The results of long term forecast error (T850 and Z500) at Day1(n=4), Day3(n=12) and Day5(n=20) for different loss type on various datasize , data resolution and model structure. The best score for each setting is highlighted in red. The L2 label signifies the usage of MSE while Ln for MLSE.

● **Setup**: We will assess the generalizability of the MLSE across various parameters, including the size and resolution of the dataset. The WeatherBench dataset is split into three parts: WeatherBench$32 \times 64$-55k, which has around 55,000 samples with a resolution of $32 \times 64$; WeatherBench$32 \times 64$-300k, which has around 330,000 samples with a resolution of $32 \times 64$; and WeatherBench$64 \times 128$-300k, which has around 330,000 samples with a resolution of $64 \times 128$. Due to the resource limitation, we have to postpone the larger resolution test into future. The MLSE trick is only applicable in the finetuning phase. Hence, all finetuning experiments were initiated with the same pretrained weights, and they all shared the same hyper-configuration. More details about the dataset and training can be found in the Supplementary. We present the results in terms

of the Root Mean Square Error (RMSE) for two weather states, T850 and Z500, following the convention in weather simulation tasks. Meanwhile, we will assess the generalizability of MLSE across different structures with two major structures: the AFNOnet, referred to as FourCastNet Pathak et al. (2022), and the Vision Transformer, referred to as Lgnet Chen et al. (2023).

• **Result**: Table 1 presents the RMSE values for T850 and Z500 at different time steps using both MSE and MLSE, with varying training data lengths at $m = 1$ (T+1), $m = 2$ (T+2), and $m = 3$ (T+3). T+1 represents the pretrained model and serves as the initial weight for the T+2 and T+3 models. This model uses the current time stamp as input and predicts the state of the next time stamp by minimizing the MSE loss. The T+2 and T+3 models are finetuned on the next-next ($m = 2$) and next-next-next ($m = 3$) time stamp datasets, respectively. Please note that during evaluation, we autoregressively generate future states up to $n = 4, 12, 20$. The time unit for our dataset is 6 hours, hence the Day1, Day3, and Day5 in real. As demonstrated in Table 1, the long-term forecasting performance is significantly enhanced by accounting for higher-order error and the application of MLSE. The results of Table 1 suggest several points: To achieve reliable long-term forecast estimation, the finetuning process (guided by next-next prediction) is essential; This effect is general and independent of the model structure and dataset properties; Incorporating more future information can guide the model towards better performance; The marginal utility of future information decays very rapidly; The effect of MLSE is orthogonal to the factors above, and it can notably catalyze the influence of future time stamps.

## 3.2 RESULTS ON TIME SERIES DATASET

In contrast, time series datasets have distinct characteristics, such as a smaller number of features and sample frequency, more pronounced periodicity, and more explicit time dependencies. Therefore, time series usually combine a sequence of timestamps into a single state to increase the flexibility of the data. Modern time-series machine learning research Wu et al. (2022) often uses an end-to-end-forecasting paradigm for long-term predictions. For instance, they input 96 timestamps and directly generate the subsequent 960 timestamps. In this paper, we find that in most cases, the pretraining-finetuning paradigm with MLSE yields better performance than the end-to-end forecasting paradigm.

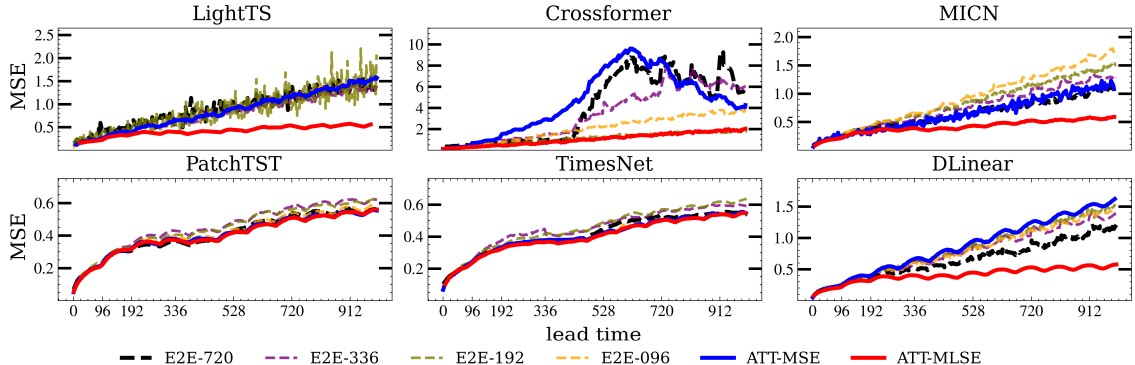

Figure 2: The long-term forecast errors for six models in Table 2 on the ETTm2 dataset are presented. The red line is the prediction result after training with ATT-MLSE, and the black line is the prediction result of end-to-end modeling $96 \times 720$. The blue line

• **Setup**: We evaluate the long-term forecasting performance of our proposed MLSE loss on six popular time series datasets ETTh1 Zhou et al. (2021), ETTh2 Zhou et al. (2021), ETTm1 Zhou et al. (2021) Zhou et al. (2021), ETTm2 Trindade (2016), Exchange Lai et al. (2018) and Weather Wet with six state-of-the-art

methods, including Crossformer Hassanin et al. (2022), DLinear Zeng et al. (2023), LightTS Zhang et al. (2022), MICN Wang et al. (2023), PatchTST Nie et al. (2022) and TimesNet Wu et al. (2022). All architectures and hyperparameters are the same as the TimesNet benchmark Github repos thuml (2023).

We will compare the performance of two prediction paradigms: the end-to-end forecast (E2E) and the autoregressive pretrain-finetune (ATT). In the E2E setting, provided by the benchmark, a fixed 96-length sequence is received as input and subsequently produces a sequence varying from 96 to 192, 336, and 720. The ATT paradigm uses the 96 to 96 E2E model for pretraining and finetunes the model for a few epochs on the next and next 96 sequence.During evaluation, all models will autoregressively generate a subsequent 960 sequence from the starting 96 input sequence. For instance, a 96 to 192 E2E model uses the input to generate stamps from 0 to 192 and employs the 0-96 portion as its input to generate stamps from 192 to 288, etc. A more complete experimental results, which contain 12 models, are provided in the Supplementary.

| model | | Crossformer | | | DLinear | | | LightTS | | | MICN | | | PatchTST | | | TimesNet | | |
|---|---|---|---|---|---|---|---|---|---|---|---|---|---|---|---|---|---|---|---|
| | | E2E | ATT | | E2E | ATT | | E2E | ATT | | E2E | ATT | | E2E | ATT | | E2E | ATT | |
| | | L2 | L2 | Ln | L2 | L2 | Ln | L2 | L2 | Ln | L2 | L2 | Ln | L2 | L2 | Ln | L2 | L2 | Ln |
| ETTh1 | 096 | 0.414 | 0.367 | 0.363 | 0.396 | 0.366 | 0.366 | 0.435 | 0.402 | 0.401 | 0.394 | 0.362 | 0.362 | 0.378 | 0.346 | 0.349 | 0.389 | 0.360 | 0.360 |
| | 192 | 0.450 | 0.418 | 0.410 | 0.445 | 0.416 | 0.413 | 0.494 | 0.443 | 0.439 | 0.454 | 0.403 | 0.403 | 0.437 | 0.381 | 0.386 | 0.437 | 0.395 | 0.397 |
| | 336 | 0.731 | 0.468 | 0.444 | 0.490 | 0.446 | 0.437 | 0.552 | 0.469 | 0.457 | 0.598 | 0.449 | 0.448 | 0.466 | 0.401 | 0.400 | 0.490 | 0.414 | 0.420 |
| | 720 | 0.609 | 0.631 | 0.552 | 0.509 | 0.529 | 0.496 | 0.613 | 0.557 | 0.521 | 0.696 | 0.620 | 0.620 | 0.507 | 0.455 | 0.426 | 0.520 | 0.462 | 0.478 |
| ETTh2 | 096 | 0.617 | 0.586 | 0.533 | 0.348 | 0.258 | 0.215 | 0.428 | 0.351 | 0.265 | 0.338 | 0.262 | 0.212 | 0.306 | 0.211 | 0.206 | 0.321 | 0.234 | 0.230 |
| | 192 | 1.585 | 0.915 | 0.830 | 0.479 | 0.362 | 0.257 | 0.582 | 0.495 | 0.311 | 0.494 | 0.353 | 0.252 | 0.377 | 0.259 | 0.248 | 0.390 | 0.271 | 0.260 |
| | 336 | 2.680 | 1.334 | 1.110 | 0.596 | 0.526 | 0.300 | 0.688 | 0.693 | 0.382 | 0.593 | 0.488 | 0.293 | 0.415 | 0.313 | 0.296 | 0.437 | 0.320 | 0.296 |
| | 720 | 3.497 | 2.017 | 1.491 | 0.825 | 1.014 | 0.371 | 1.006 | 1.118 | 0.586 | 0.827 | 0.852 | 0.369 | 0.449 | 0.398 | 0.365 | 0.487 | 0.388 | 0.357 |
| ETTm1 | 096 | 0.374 | 0.364 | 0.340 | 0.345 | 0.327 | 0.331 | 0.402 | 0.376 | 0.376 | 0.322 | 0.313 | 0.306 | 0.323 | 0.306 | 0.306 | 0.336 | 0.332 | 0.327 |
| | 192 | 0.448 | 0.444 | 0.415 | 0.382 | 0.370 | 0.371 | 0.431 | 0.420 | 0.421 | 0.358 | 0.364 | 0.354 | 0.369 | 0.347 | 0.345 | 0.385 | 0.374 | 0.364 |
| | 336 | 0.715 | 0.541 | 0.506 | 0.414 | 0.428 | 0.420 | 0.466 | 0.474 | 0.475 | 0.393 | 0.437 | 0.415 | 0.399 | 0.398 | 0.387 | 0.415 | 0.428 | 0.406 |
| | 720 | 0.687 | 0.710 | 0.728 | 0.472 | 0.547 | 0.515 | 0.561 | 0.582 | 0.576 | 0.506 | 0.624 | 0.588 | 0.459 | 0.495 | 0.468 | 0.476 | 0.528 | 0.497 |
| ETTm2 | 096 | 0.284 | 0.296 | 0.216 | 0.190 | 0.194 | 0.184 | 0.208 | 0.211 | 0.197 | 0.190 | 0.195 | 0.176 | 0.178 | 0.184 | 0.182 | 0.189 | 0.191 | 0.192 |
| | 192 | 0.374 | 0.500 | 0.298 | 0.275 | 0.280 | 0.252 | 0.317 | 0.299 | 0.262 | 0.270 | 0.262 | 0.243 | 0.241 | 0.250 | 0.248 | 0.251 | 0.254 | 0.252 |
| | 336 | 0.803 | 0.785 | 0.414 | 0.367 | 0.389 | 0.318 | 0.383 | 0.414 | 0.325 | 0.355 | 0.333 | 0.306 | 0.321 | 0.313 | 0.311 | 0.328 | 0.315 | 0.310 |
| | 720 | 3.200 | 1.570 | 0.774 | 0.550 | 0.660 | 0.421 | 0.778 | 0.705 | 0.425 | 0.535 | 0.528 | 0.409 | 0.401 | 0.415 | 0.409 | 0.422 | 0.414 | 0.409 |
| Exchange | 096 | 0.288 | 0.461 | 0.391 | 0.094 | 0.095 | 0.097 | 0.141 | 0.165 | 0.103 | 0.092 | 0.097 | 0.091 | 0.084 | 0.078 | 0.079 | 0.102 | 0.104 | 0.104 |
| | 192 | 0.661 | 0.635 | 0.488 | 0.185 | 0.190 | 0.163 | 0.310 | 0.378 | 0.196 | 0.184 | 0.186 | 0.153 | 0.184 | 0.145 | 0.145 | 0.211 | 0.184 | 0.184 |
| | 336 | 1.288 | 0.803 | 0.572 | 0.342 | 0.337 | 0.225 | 0.478 | 0.655 | 0.320 | 0.327 | 0.318 | 0.214 | 0.347 | 0.216 | 0.215 | 0.346 | 0.265 | 0.265 |
| | 720 | 1.708 | 1.029 | 0.730 | 0.747 | 0.592 | 0.512 | 0.937 | 0.942 | 0.594 | 0.795 | 0.565 | 0.453 | 0.847 | 0.613 | 0.606 | 0.903 | 0.706 | 0.704 |
| weather | 096 | 0.175 | 0.172 | 0.165 | 0.196 | 0.188 | 0.188 | 0.173 | 0.168 | 0.167 | 0.183 | 0.184 | 0.184 | 0.175 | 0.171 | 0.175 | 0.169 | 0.166 | 0.165 |
| | 192 | 0.232 | 0.215 | 0.204 | 0.235 | 0.225 | 0.226 | 0.234 | 0.206 | 0.200 | 0.239 | 0.216 | 0.217 | 0.222 | 0.213 | 0.217 | 0.228 | 0.208 | 0.207 |
| | 336 | 0.277 | 0.266 | 0.243 | 0.282 | 0.268 | 0.272 | 0.266 | 0.250 | 0.237 | 0.275 | 0.254 | 0.250 | 0.279 | 0.260 | 0.264 | 0.283 | 0.254 | 0.254 |
| | 720 | 0.368 | 0.363 | 0.305 | 0.344 | 0.358 | 0.349 | 0.344 | 0.338 | 0.296 | 0.349 | 0.329 | 0.311 | 0.355 | 0.345 | 0.350 | 0.354 | 0.340 | 0.341 |

Table 2: The table presents the forecast performance (MSE) of the end-to-end-forecast (E2E) and autoregressive pretrain-finetune (ATT) paradigms across various datasets and models. 'L2' and 'Ln' denote the training losses. Red and blue highlights show where ATT outperforms or underperforms E2E, respectively. In each row, underlined values and blue backgrounds respectively signify the best scores for current model and across all models. The L2 label signifies the usage of MSE while Ln for MLSE.

• **Result**: Table 2 depict the ATT benefits the majority (83%) of tasks, as indicated by the predominance of red figures. This suggests that while end-to-end (E2E) methods mitigate the error accumulation issue, they still struggle to model long-term temporal relationships effectively. However, the ATT approach alleviates this problem. Furthermore, our proposed MLSE loss significantly boosts autoregressive performance, with approximately 76% of cases outperforming the baseline MSE loss. For longer forecasts, such as sequences of 336 and 720, MLSE outperforms other settings in 10 out of 12 cases, while the remaining cases demonstrate very close performance to E2E. Figure 2 shows the full prediction error curves from stamp 0 to stamp 960 for six models in the ETTm2 data set, illustrating the advantage of our proposed MLSE loss. It shows that the ATT-MSE method (blue line) may not always improve long-term forecast performance, potentially explaining

why previous literature has not focused on the ATT paradigm. However, simply adding a logarithmic operation (red line) enhances all models' long-term forecasting ability to state-of-the-art levels.

## 4 SUPPRESSING ERROR ACCUMULATION

This section introduces our analysis for modeling long-term forecast error and deriving the logarithmic trick. Although many other proposals can be considered based on this analysis, our goal is to find the most efficient solution. After thorough a comprehensive studies, we identify MLSE as the best candidate due to its zero-cost and plug-and-play nature.

Our analysis begins with this consideration: traditional ATT methods directly back-propagate on the higher-order loss, $\varepsilon_{t+n}^N$, to optimize long-term future error. However, noting that we employ recurrent forward processing and compute the average error throughout the evaluation test, $\mathcal{E}^N = \frac{1}{N}\sum_t \varepsilon_{t+n}^N$. Thus, the effects of last-order error ($\mathcal{E}^{N-1}$) and the next-order error ($\mathcal{E}^N$) are not fully orthogonal. In fact, we can observe that the long-term forecast performance significantly improves after including the second-order error, and continues to improve with the addition of the third- and fourth-order terms, albeit with diminishing returns. This observation motivates us to model the relationship between different order errors and find the most effective way to incorporate the information from higher-order losses.

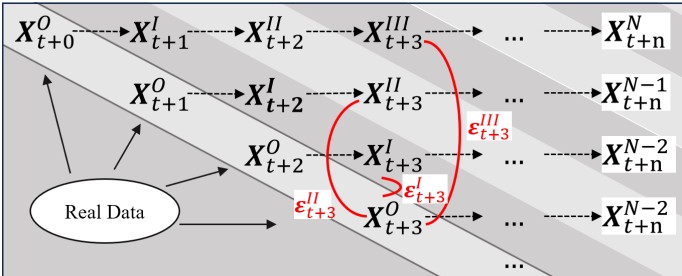

Figure 3: The diagram for the advanced autoregressive graph.

Fig.3 extends the autoregressive forward graph from Fig.1. We now consider predictions from the subsequent true timestamp $X_{t+2}^I$ and onward, as well as their corresponding errors relative to the ground truth. By incorporating the vector-error term $\varepsilon_{t+n}^N$, we can construct an equivalent training process:

$$X_{t+N}^O = X_{t+N}^I + \varepsilon_{t+N}^I = X_{t+N}^{II} + \varepsilon_{t+N}^{II} = \cdots = X_{t+N}^N + \varepsilon_{t+N}^N$$

Notice that in the pretrain phase, we compute and optimize 1st-order error; In the finetune phase, we involve higher-order term. Therefore, we consider the first-order shift of order $N$:

$$\varepsilon_{t+N}^N = \varepsilon_{t+N}^I + (X_{t+N}^I - X_{t+N}^N) = \varepsilon_{t+N}^I + M_{t+N-1}^{N-1}\varepsilon_{t+N-1}^{N-1}$$

where $\varepsilon_{t+N}^N$ is the $N$-order error and $\varepsilon_{t+N}^I$ is the first order error at time $t+N$,. The $M_{t+N-1}^{N-1}$ is the propagation matrix produced by the Jacobian operator followed the multi-dimension Mean Value Theorem:

$$X_{t+N}^I - X_{t+N}^N = f_\theta(X_{t+N-1}^O) - f_\theta(X_{t+N-1}^{N-1}) = \nabla f_\theta(X_\delta)(X_{t+N-1}^O - X_{t+N-1}^{N-1}) = M_{t+N-1}^{N-1}\varepsilon_{t+N-1}^{N-1}.$$

where $X_\delta$ is a state between $X_{t+N-1}^O$ and $X_{t+N-1}^{N-1}$.

To address the final error, we take the norm-square $||v|| = \sum v_s^2$ for each vector,

$$||\varepsilon_{t+N}^N|| = ||\varepsilon_{t+N}^I + M_{t+N-1}^{N-1}\varepsilon_{t+N-1}^{N-1}|| \approx ||\varepsilon_{t+N}^I|| + ||M_{t+N-1}^{N-1}\varepsilon_{t+N-1}^{N-1}||$$

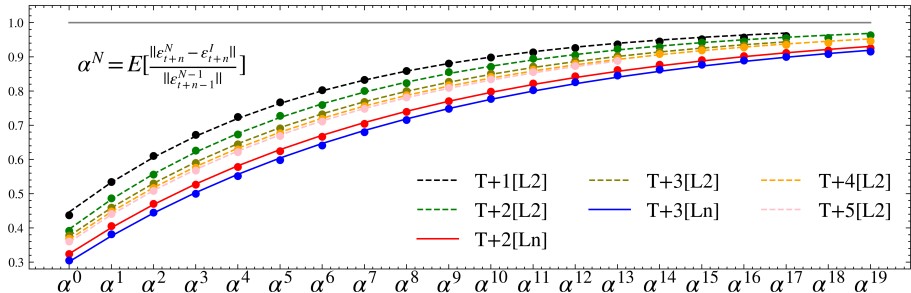

Figure 4: The figure displays the amplifier, computed using Equation 4, under different finetuning conditions. The T+1[L2] (represented in black) pertains to the pretrained weight from which all other finetune originate, and it possesses a significantly large alpha value. As more order loss per stamp are added (from T+2 to T+5), a noticeable decrease is observed as more future information is incorporated. Conversely, the inclusion of a zero-cost logarithmic term makes it possible to suppress the amplifier to a very low phase, only use small order loss. The L2 label signifies the usage of MSE while Ln for MLSE.

Typically, the dimension of the state ($K$) is quite large, which results in $\varepsilon_{t+N}^I$ and $M_{t+N-1}^{N-1}\varepsilon_{t+N-1}^{N-1}$ "almost" orthogonal: $\langle \varepsilon_{t+N-1}^I | M_{t+N-1}^{N-1}\varepsilon_{t+N-1}^{N-1}\rangle \approx 0$. Thus, the in-equivalent become equivalent.

We define the "error amplifier"

$$\alpha_t^{N-1} = \frac{||M_{t+N-1}^{N-1}\varepsilon_{t+N-1}^{N-1}||}{||\varepsilon_{t+N-1}^{N-1}||} = \frac{||X_{t+N}^N - X_{t+N}^I||}{||X_{t+N-1}^{N-1} - X_{t+N-1}^O||} = \frac{||\varepsilon_{t+N}^N - \varepsilon_{t+N}^I||}{||\varepsilon_{t+N-1}^{N-1}||} \approx \frac{||\varepsilon_{t+N}^N|| - ||\varepsilon_{t+N}^I||}{||\varepsilon_{t+N-1}^{N-1}||} \tag{2}$$

And define $\mathcal{E}^N \equiv E(\mathcal{E}_t^N) \equiv E(||\varepsilon_{t+N}^N||)$ refers to the long term forecast L2 error we evaluate in a series of tasks, we can obtain a very straightforward relationship between the next error and the current error.

$$\mathcal{E}^N = \mathcal{E}^I + \alpha^{N-1}\mathcal{E}^{N-1} = (1 + \alpha^N + \alpha^N\alpha^{N-1} + \alpha^N\alpha^{N-1}\alpha^{N-2} + \dots)\mathcal{E}^I \tag{3}$$

where we make the independent assumption and decompose the expected product $E[\alpha_t^N\mathcal{E}_{t+N}^N] = E[\alpha_t^N]E[\mathcal{E}_{t+N}^N] = \alpha^N E[\mathcal{E}_{t+N}^N]$.

Therefore, the most efficient method of minimizing the long-term forecast error $\mathcal{E}^N$ is to optimize the first-order error $\mathcal{E}^I$ and the error amplifier $\alpha^N$. In Figure4, we plot the amplifier $\alpha$ from $\alpha^1$ to $\alpha^20$ for various pretraining-finetuning strategies. It is evident that the amplifier decreases as more order losses are incorporated in the standard ATT-MSE strategy. However, the ATT-MLSE method can significantly expedite this process, making it possible to achieve very low long-term forecast errors using only a small amount of order information.

$$\min\{\mathcal{E}^I, \alpha^2, \alpha^3, \cdots\}$$

In the pretrain-finetune paradigm, the error function $\mathcal{E}^I$ is already optimized during the pretraining phase. Subsequently, during the finetuning phase, we aim to optimize the error amplifier $\alpha^N$. However, directly computing $\alpha^N$ poses a significant challenge. From definition 4, there are two methods to derive $\alpha$: 1) Calculate the Jacobin-vector-product $M_t^N\varepsilon_t^N$ directly. Unfortunately, this approach requires huge computing resources in the current auto-differentiation engineering landscape. 2) Compute the extra state $X_{t+N}^I = f_\theta(X_{t+N-1}^O)$. For a sequence of data points $X_t^O, X_{t+2}^O, \cdots, X_{t+n}^O$, we need not only to store the recurrent prediction $X_{t+1}^N, X_{t+2}^{II}, \cdots, X_{t+n}^N$ but also preserve all the first-order results $X_{t+2}^I, \cdots, X_{t+n}^I$. This requirement introduces new memory challenges for large-scale systems. Furthermore, we employ a mini-batch strategy

for weight updates. It is under investigation that simultaneously do back-propagation for the batch error $||\boldsymbol{X}_{t+N}^N - \boldsymbol{X}_{t+N}^O||$ and batch of error division $\alpha^N = ||\boldsymbol{X}_{t+N}^N - \boldsymbol{X}_{t+N}^I||/||\boldsymbol{X}_{t+N-1}^{N-1} - \boldsymbol{X}_{t+N-1}^O||$. Moreover, the division operation to compute $\alpha$ will lead to numerical instability during the train.

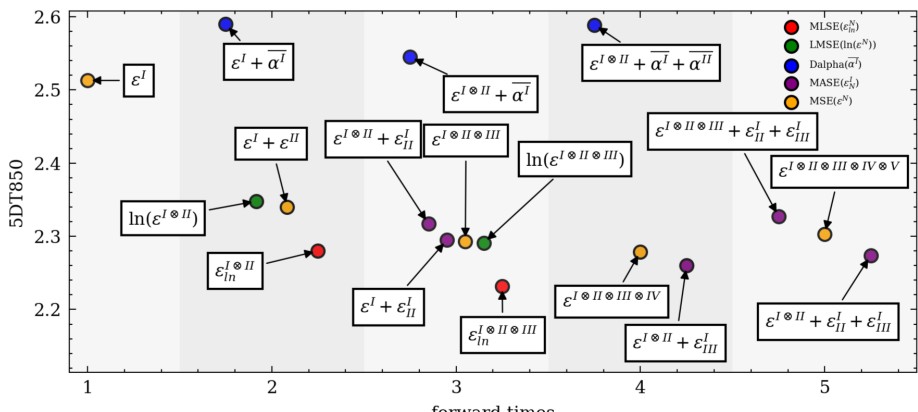

Figure 5: The figure shows the RMSE in terms of T850 at Day5(n=20) for different setting. The orange node represents the normal settings that only use MSE loss. LMSE loss(green) shows no advantage. Dalpha(blue) result in bad training stability. MASE(purple) benefits from amplifier design but require one more forward resource. MLSE(red) providing best performance without extra computational cost. The symbol $\otimes$ signifies a sum of order-based quantities, e.g., $\varepsilon^{I \otimes II} \equiv \varepsilon^I + \varepsilon^{II}$ denotes train on $T+1$ and $T+2$ normal MSE loss.

Therefore, we propose several candidates to identify the optimal method for guiding the model learning of the amplifier information. There are several key rules to ensure numerical stability: To circumvent numerical issues caused by division, we apply a logarithmic wrapper. Although logarithms carry their own numerical challenges, these can be more easily mitigated by adding a small epsilon value. When combining multiple losses, we should avoid negative coefficients.

In Figure 5, we test not only the 5 Days T850 RMSE of MSE-finetune and MLSE-finetune configurations, but also the LMSE, MASE, and Dalpha finetune strategies. The Dalpha strategy directly minimizes the error amplifier $\alpha_t^N = ||\boldsymbol{X}_{t+N}^N - \boldsymbol{X}_{t+N}^I||/||\boldsymbol{X}_{t+N-1}^{N-1} - \boldsymbol{X}_{t+N-1}^O||$. It suffers heavily from the division operation. As an improved alternative, we propose the MASE strategy, defined as MASE$=\sum_N \sum_t [||\boldsymbol{X}_{t+N}^N - \boldsymbol{X}_{t+N}^I|| + ||\boldsymbol{X}_{t+N-1}^I - \boldsymbol{X}_{t+N-1}^O|||]$. Original, it should be $\log(||\boldsymbol{X}_{t+N}^N - \boldsymbol{X}_{t+N}^I||) + \log(||\boldsymbol{X}_{t+N}^N - \boldsymbol{X}_{t+N}^I||)$. Applying a logarithm outside the average operation is equivalent to introducing a dynamic coefficient to the gradient and would be mitigated by using a momentum-based optimizer like Adam. Take the LMSE attempts as an example, it computes $\alpha^N \approx E[(||\varepsilon_{t+N}^N|| - ||\varepsilon_{t+N}^I||)/||\varepsilon_{t+N-1}^{N-1}||] \approx (\mathcal{E}^N - \mathcal{E}^I)/\mathcal{E}^{N-1}$. When $n = 2$, it minimizes $\{\alpha^{II} = \mathcal{E}^{II}/\mathcal{E}^I - 1, \mathcal{E}^I\}$, then is converted into $\ln[\mathcal{E}^{II}] - \ln[\mathcal{E}^I]$ and $\ln[\mathcal{E}^I]$. We use the coefficients with 1:2 and achieve: LMSE $=\sum_N \sum_t \ln[\sum_s (\varepsilon_{t+N,s}^N)^2]$. The experiment shows that such a operation won't take any advantage. Therefore, we choose to either remove the external logarithm or move the logarithm inside the average operation. This optimizes the geometric lower boundary of the term $\ln(||\frac{1}{K}\sum_s \varepsilon_{t+2,s}^{II}||) \geq \frac{1}{K}\sum_s (\ln||\varepsilon_{t+2}^{II}||)$. This transformation changes the LMSE into MLSE$=\sum_N \sum_t \sum_s \ln[(\epsilon_{t+N,s}^N)^2]$.

Logarithmic forms can pose numerical challenges when the input is close to 0. However, these drawbacks can be easily mitigated by adding a small epsilon value. In our experiment, we set the epsilon value as a hyperparameter and test the validation of MLSE from small eps to large eps.

| | model | FourCastNet | | | | LgNet | | | | |
|---|---|---|---|---|---|---|---|---|---|---|
| | eps | L:1e-2 | L:1e-3 | L:1e-5 | MSE | L:1 | L:1e-2 | L:1e-3 | L:1e-5 | MSE |
| T850 | Day 1 | 0.99 | 0.99 | 1.01 | 1.08 | 0.86 | 0.83 | 0.84 | 0.85 | 0.88 |
| | Day 3 | 1.49 | 1.50 | 1.52 | 1.60 | 1.35 | 1.31 | 1.31 | 1.34 | 1.38 |
| | Day 5 | 2.25 | 2.25 | 2.28 | 2.34 | 2.06 | 2.02 | 2.03 | 2.05 | 2.11 |
| Z500 | Day 1 | 70 | 69 | 70 | 81 | 61 | 57 | 56 | 60 | 63 |
| | Day 3 | 209 | 209 | 213 | 226 | 179 | 172 | 170 | 177 | 183 |
| | Day 5 | 418 | 415 | 422 | 438 | 367 | 357 | 355 | 361 | 378 |

Table 3: The results of RMSE in terms of T850 and Z500 at different time steps for FourCastNet and LgNet by using various eps.

Table 3 shows that, irrespective of the epsilon (eps) value, even set to 1, the use of the logarithmic form consistently results in noticeable improvements. This indicates that the logarithmic form is essential. Interestingly, optimal performance is not achieved with the smallest epsilon value. In this study, we treat epsilon as a hyperparameter and fix it at $1e-5$ for all experiments.

## 5    LIMITATION

Although we have demonstrated that ATT+MLSE can bring significant improvements to long-term performance on many systems, there are several issues that require further exploration.

- The efficacy of ATT-MLSE is dependent on the temporal characteristics of the task. Only tasks that are theoretically capable of making long-term forecasts, such as self-evolution, can benefit from them. For example, it fails for the driving video prediction task. A pure video task does not include global map information, thus it lacks the context to predict whether there is a person on the street based solely on its own recurrent information. The time series traffic dataset, as shown in Supplementary, serves as yet another example.

- There exists many other views for analysis. From a loss objectives viewpoint, MLSE shares a close mathematical resemblance with p-SNR, except that p-SNR serves as a discrete digital metric. In terms of multiclass optimization, MLSE employs a "winner-takes-all" strategy, allocating more bias to smaller loss components. Readers may draw parallels between MLSE and Uncertainty Loss Kendall et al. (2018), where element-wise variance is trained to balance each component's contributions, resulting in a similar "winner-takes-all" approach. However, it has been observed that MLSE is effective only when applied to both order-$N$ and order-1; applying it solely to order-1 does not produce any significant impact. This observation implies that factors beyond the "loss math" and "component balance" directly influence the error behavior. A more comprehensive understanding of these factors requires in further research.

## 6    CONCLUSION

In this paper, we introduce the logarithm trick named Mean Log Square Error (MLSE), a plug-and-play and zero-cost modification to the standard MSE loss function, and greatly enhances long-term forecasting performance without necessitating any extra computation. The efficacy of MLSE was confirmed through extensive experiments on various model architectures and datasets. We proposed a phenomenological theory to analyze the feasibility and limitations of MLSE by modeling the rate of error accumulation. To the best of our knowledge, such an angle of analysis has rarely been explored in this area. While our theory is not perfect, we hope it can offer a novel viewpoint for improving the AI's ability in long-term forecasting tasks, such as weather forecasting, financial analysis, field dynamic evolution, and more.

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
