# SUPPLEMENTARY: THE LOGARITHM TRICK: ACHIEVE BETTER LONG TERM FORECAST VIA MEAN LOGARITHM SQUARE LOSS

## 1 PHENOMENOLOGICAL THEORY

A typical Autoregressive training task (ATT) consists of a model $f_\theta : \mathbb{R}^d \to \mathbb{R}^d$ and a sequence of unanimous data

$$\vec{X}_t, \vec{X}_{t+1}, \ldots, \vec{X}_{t+N},$$

where each $\vec{X}_t = [\boldsymbol{X}_{t,1}, \boldsymbol{X}_{t,2}, \cdots, \boldsymbol{X}_{t,s}]$ represents a time snapshot of the state at step $t$ with $s$ degrees of freedom. For example, a full resolution WeatherBenchsnapshot $\vec{X}_t$ is a large 4D tensor with dimensions $s = 5 \times 37 \times 720 \times 1440 \approx 2 \times 10^8$; A typical time series prediction task such as ETTh1consist of series of $1 \times 7$ vector. When the input is a sequence of 96 time stamp, the freedom is viewed as $s = 96 \times 7 = 672$.

An ideal model $f$ is expected to reproduce any $\vec{X}_{t+n+1}$ by $f(\vec{X}_{t+n})$

$$\vec{X}_t \xrightarrow{f} \vec{X}_{t+1} \xrightarrow{f} \vec{X}_{t+2} \xrightarrow{f} \vec{X}_{t+3} \xrightarrow{f} \cdots \xrightarrow{f} \vec{X}_{t+N}$$

However, in most tasks, it is impossible to find the ideal model. Therefore, we need to use approximation functions such as neural networks to simulate this mapping, and the ideal model degenerates to a parameterized function $f \to f_\theta$. The task becomes an optimization problem of order $N$:

$$\min \sum_t |\vec{X}_{t+N} - \underbrace{f f \cdots f}_{N}(\vec{X}_t)| = \min \sum_t \mathcal{E}_t^N = \min \sum_t \mathcal{E}_{t,s}^N$$

Researchers usually back-propagate on the first low-order errors. For example, in the FourCastNetPathak et al. (2022) forwards twice on the "fine-tune" phase and Pangu only deal with order-1 errors. Several factors are concerned: Firstly, achieving the ideal model on the order-1 loss $\mathcal{E}_{\forall t}^1 \to 0$ directly implies that $\mathcal{E}_{\forall t}^N \to 0$. Secondly, training on $\mathcal{E}_t^N$ requires $N$ times forward-prediction and back-propagation, making it quite unaffordable in large system simulations compared to single loss optimization. Thirdly, the marginal benefit of increasing $N$ decays rapidly in experience, and there must be a trade-off between accuracy and speed.

Thus, a comprehensive ATT optimization task of order $N$ is formalized as

$$\min \mathcal{E} = \sum_t (\mathcal{E}_t^1 + \mathcal{E}_t^2 + \cdots + \mathcal{E}_t^N) = \sum_N \sum_t \mathcal{E}_t^N = \sum_N \sum_t \sum_s \mathcal{E}_{t,s}^N$$

We expect a model trained on order $N$ to be capable of forecasting a longer future beyond $N$, as the physical world is believed to have localized correlations, and a machine learning model trained on neighboring relationships should be able to extrapolate to a further distance. This pipeline has been validated in many tasksBi et al. (2022); Lam et al. (2022); Chen et al. (2023), which have significantly outperformed traditional methods.

For any step $t$, its time snapshot $\boldsymbol{X}_t$ and a differentiable and smooth parameter neural function $f_\theta$, we will not only obtain a snapshot chain

$$
\begin{array}{ccccccccccc}
\boldsymbol{X}_t^O & \xrightarrow{f_\theta} & \boldsymbol{X}_{t+1}^{\mathrm{I}} & \xrightarrow{f_\theta} & \boldsymbol{X}_{t+2}^{\mathrm{II}} & \xrightarrow{f_\theta} & \boldsymbol{X}_{t+3}^{\mathrm{III}} & \xrightarrow{f_\theta} & \cdots & \xrightarrow{f_\theta} & \boldsymbol{X}_{t+N}^{N} \\
\Updownarrow & & \Updownarrow & & \Updownarrow & & \Updownarrow & & & & \Updownarrow \\
\boldsymbol{X}_t^O & & \boldsymbol{X}_{t+1}^O & & \boldsymbol{X}_{t+2}^O & & \boldsymbol{X}_{t+3}^O & & \cdots & & \boldsymbol{X}_{t+N}^O
\end{array}
$$

but also a snapshot tree. lets take $N = 5$ as an example:

$$
\begin{pmatrix}
\boldsymbol{X}_t^O & \boldsymbol{X}_{t+1}^{\mathrm{I}} & \boldsymbol{X}_{t+2}^{\mathrm{II}} & \boldsymbol{X}_{t+3}^{\mathrm{III}} & \boldsymbol{X}_{t+4}^{\mathrm{IV}} & \boldsymbol{X}_{t+5}^{V} \\
 & \boldsymbol{X}_{t+1}^O & \boldsymbol{X}_{t+2}^{\mathrm{I}} & \boldsymbol{X}_{t+3}^{\mathrm{II}} & \boldsymbol{X}_{t+4}^{\mathrm{IV}} & \boldsymbol{X}_{t+5}^{\mathrm{IV}} \\
 & & \boldsymbol{X}_{t+2}^O & \boldsymbol{X}_{t+3}^{\mathrm{I}} & \boldsymbol{X}_{t+4}^{\mathrm{II}} & \boldsymbol{X}_{t+5}^{\mathrm{III}} \\
 & & & \boldsymbol{X}_{t+3}^O & \boldsymbol{X}_{t+4}^{\mathrm{I}} & \boldsymbol{X}_{t+5}^{\mathrm{II}} \\
 & & & & \boldsymbol{X}_{t+4}^O & \boldsymbol{X}_{t+5}^{\mathrm{I}} \\
 & & & & & \boldsymbol{X}_{t+5}^O
\end{pmatrix}
\tag{1}
$$

In our notation, the superscript represents the order order. For instance, $O$ refers to the real data, while $I$ represents the data generated by performing one forward pass from a ground truth through the function $f$. The subscript indicates the time stamp of the prediction. It is important to note that only tensors with the same time stamp can be compared with each other. As a result, we can categorize all errors according to their order by follow identity:

$$
\begin{aligned}
\boldsymbol{X}_{t+1}^O &= \boldsymbol{X}_{t+1}^{\mathrm{I}} + \varepsilon_{t+1}^{\mathrm{I}} \\
\boldsymbol{X}_{t+2}^O &= \boldsymbol{X}_{t+2}^{\mathrm{I}} + \varepsilon_{t+2}^{\mathrm{I}} = \boldsymbol{X}_{t+2}^{\mathrm{II}} + \varepsilon_{t+2}^{\mathrm{II}} \\
\boldsymbol{X}_{t+3}^O &= \boldsymbol{X}_{t+3}^{\mathrm{I}} + \varepsilon_{t+3}^{\mathrm{I}} = \boldsymbol{X}_{t+3}^{\mathrm{II}} + \varepsilon_{t+3}^{\mathrm{II}} = \boldsymbol{X}_{t+3}^{\mathrm{III}} + \varepsilon_{t+3}^{\mathrm{III}} \\
&\cdots
\end{aligned}
$$

when the order order meets the smae time stamp, the error is just the original error we want to minimized. $\mathcal{E}_t^1 = ||\varepsilon_{t+1}^{\mathrm{I}}||, \mathcal{E}_t^2 = ||\varepsilon_{t+2}^{\mathrm{II}}||, \ldots$

Those identity reveal the realation between low order error and its higher order partner. Start from $N = 2$, we can achieve

$$
\begin{aligned}
\boldsymbol{X}_{t+2}^O &= \boldsymbol{X}_{t+2}^{\mathrm{I}} + \varepsilon_{t+2}^{\mathrm{I}} = \boldsymbol{X}_{t+2}^{\mathrm{II}} + \varepsilon_{t+2}^{\mathrm{II}} \\
&\rightarrow f(\boldsymbol{X}_{t+1}^O) + \varepsilon_{t+2}^{\mathrm{I}} = f(\boldsymbol{X}_{t+1}^{\mathrm{I}}) + \varepsilon_{t+2}^{\mathrm{II}} \\
&\rightarrow f(\boldsymbol{X}_{t+1}^O) - f(\boldsymbol{X}_{t+1}^{\mathrm{I}}) + \varepsilon_{t+2}^{\mathrm{I}} = \varepsilon_{t+2}^{\mathrm{II}} \\
&\rightarrow \nabla f(\boldsymbol{X}_\delta)(\boldsymbol{X}_{t+1}^O - \boldsymbol{X}_{t+1}^{\mathrm{I}}) + \varepsilon_{t+2}^{\mathrm{I}} = \varepsilon_{t+2}^{\mathrm{II}} \\
&\rightarrow \nabla f(\boldsymbol{X}_\delta)(\varepsilon_{t+1}^{\mathrm{I}}) + \varepsilon_{t+2}^{\mathrm{I}} = \varepsilon_{t+2}^{\mathrm{II}} \\
&\rightarrow M_{t+1}^{\mathrm{I}}(\varepsilon_{t+1}^{\mathrm{I}}) + \varepsilon_{t+2}^{\mathrm{I}} = \varepsilon_{t+2}^{\mathrm{II}}
\end{aligned}
$$

Here, we utilize the Mean Value Theorem of multi-dimensional calculus, where $\boldsymbol{X}_\delta \in [\boldsymbol{X}_{t+1}^O, \boldsymbol{X}_{t+1}^{\mathrm{I}}]$ and $\nabla f$ represents the Jacobian operator. This operator produces a matrix $M_{t+1}^{\mathrm{I}}(\delta)$ for $\boldsymbol{X}_\delta$, which is abbreviated as $M_{t+1}^{\mathrm{I}}$. The metric $\delta$ is dependent on the model parameter $\theta$ and fluctuate between the predicted value $\boldsymbol{X}_{t+1}^{\mathrm{I}}$ and the ground truth $\boldsymbol{X}_{t+1}^O$. As $\delta \to 0$, the model $f_\theta$ approaches the ideal model $f_\theta \to f$ and can accurately estimate any $\boldsymbol{X}_\delta = \boldsymbol{X}_{t+1}^O = \boldsymbol{X}_{t+1}^{\mathrm{I}} = f(\boldsymbol{X}_t^O)$.

Similarly, any order propagation can be expressed as follows:

$$
\varepsilon_{t+N}^N = \varepsilon_{t+N}^{\mathrm{I}} + M_{t+N-1}^{N-1}\varepsilon_{t+N-1}^{N-1}
$$

where $\varepsilon^N_{t+N}$ is the $N$-order error at time $t+N$, $\varepsilon^{\mathrm{I}}_{t+N-1}$ is the first-order error at time $t+N-1$, and $M^{N-1}_{t+N-1}$ is the propagation matrix. We can observe that this is a recursive formulation that goes from $\varepsilon^{N-1}_{t+N-1}$ to $\varepsilon^N_{t+N}$.

Since all $\epsilon$ terms represent the prediction's error, they are expected to behave like high-dimensional, non-correlated, and time-independent noise fluctuation. Therefore, when the number of freedoms in $\epsilon$ is large enough, the elements from different time stamps can be considered "almost" orthogonal: $\langle \varepsilon^{\mathrm{I}}_{t+N-1} | M^{N-1}_{t+N-1} \varepsilon^{N-1}_{t+N-1} \rangle = 0$. Then the size of error for each sample is measured as

$$||\epsilon^N_{t+N}|| \approx ||\varepsilon^{\mathrm{I}}_{t+N}|| + ||M^{N-1}_{t+N-1} \varepsilon^{N-1}_{t+N-1}||$$

And the average error is measured as

$$\begin{aligned}
\mathcal{E}^N &= \frac{1}{n} \sum_t ||\epsilon^N_{t+N}|| \\
&= E[||\epsilon^{\mathrm{I}}_{t+N}||] + E[||M^{N-1}_{t+N-1} \varepsilon^{N-1}_{t+N-1}||] \\
&= \mathcal{E}^{\mathrm{I}} + E[\alpha^N_t ||\varepsilon^{N-1}_{t+N-1}||] \\
&= \mathcal{E}^{\mathrm{I}} + E[\alpha^N_t] E[||\varepsilon^{N-1}_{t+N-1}||] \\
&= \mathcal{E}^{\mathrm{I}} + \alpha^N \mathcal{E}^N
\end{aligned}$$

It is important to note that $E[\alpha^N_t] E[||\varepsilon^{N-1}_{t+N-1}||]$ may not be equal to $E[\alpha^N_t] E[||\varepsilon^{N-1}_{t+N-1}||]$ since there is no guarantee that the random variables $\alpha^N_t$ and $||\varepsilon^{N-1}_{t+N-1}||$ are independent. Geometrically, $\alpha^N_t$ is a factor that amplifies any potential fluctuations of the input $X_{t+N-1} + \varepsilon^{N-1}_{t+N-1}$ on the result. Since different $t$ corresponds to different orientations in variable space, and our model has no anisotropic prior, such an amplifier should also be "almost" independent of the orientation. Therefore, we assume that $\alpha^N_t = ||M^{N-1}_{t+N-1} \varepsilon^{N-1}_{t+N-1}|| / ||\varepsilon^{N-1}_{t+N-1}||$ is a number that only depends on the order $N$ and the model $\theta$.

Finally, we get the error propagation law of ATT problem when 1-order prediction error $\delta$ as small as enough $\delta \to 0$:

$$\mathcal{E}^N = (1 + \alpha^N + \alpha^N \alpha^{N-1} + \alpha^N \alpha^{N-1} \alpha^{N-2} + \dots) \mathcal{E}^{\mathrm{I}} \tag{2}$$

Furthermore, experiments in Sec.2 reveal that $\alpha^{i:1 \to N}$ follow an exponential decay law:

$$\alpha^N = 1 - \beta_1 \exp(-\beta_2 * N)$$

This finding implies that optimize $\alpha^1$ and $\alpha^2$ will spontaneously suppress all $\alpha^{i>2}$. Moreover, we can estimate long-range performance by computing only the first three errors $\mathcal{E}^{\mathrm{III}}$, $\mathcal{E}^{\mathrm{II}}$, and $\mathcal{E}^{\mathrm{I}}$, using them to compute $\alpha^1$ and $\alpha^2$, and fitting all $\alpha^N$ via this empirical formula. This allows us to quickly estimate the long-range prediction error $\mathcal{E}^{\mathrm{N}}$."

According to Equ.1, to optimize a model with minimum $N$-order error $\mathcal{E}^N$, we need to optimize both the 1-order error $\mathcal{E}^1$ and the error amplifier $\alpha_{i:1 \to N}$. The traditional ATT loss directly optimizes $\mathcal{E}^1 + \mathcal{E}^2$, which is actually a coupled version of this view. For example, when $N = 2$, the traditional loss configuration treats the 2-order error and 1-order error equally, leading to:

$$\mathcal{E} = \mathcal{E}^{\mathrm{II}} + \mathcal{E}^{\mathrm{I}} = \sum_t \sum_s (\varepsilon^{\mathrm{II}}_{t+2,s})^2 + \sum_t \sum_s (\varepsilon^{\mathrm{I}}_{t+1,s})^2$$

This enforces that both errors are minimized rather than optimizing the amplifier $\alpha_2 = \frac{\mathcal{E}^{\mathrm{II}}}{\mathcal{E}^{\mathrm{I}}} - 1$. However, this approach risks amplifying $\frac{\mathcal{E}^{\mathrm{II}}}{\mathcal{E}^{\mathrm{I}}}$ when both $\mathcal{E}^{\mathrm{II}}$ and $\mathcal{E}^{\mathrm{I}}$ are minimized, which can lead to very poor long-term forecasts.

A good long-term forecast trainer must decouple the errors $\mathcal{E}^N, \mathcal{E}^{N-1}, \ldots, \mathcal{E}^I$. One approach to achieve this is to directly optimize $\mathcal{E} = c_1 \frac{\mathcal{E}^{II}}{\mathcal{E}^I} + c_2 \mathcal{E}^I$. However, the reciprocal loss makes training unstable. An alternative is to use the logarithm of the mean square error, $Log\mathcal{E} = \ln \mathcal{E}^{II} + \ln \mathcal{E}^I$. Unfortunately, such a loss is not additive during mini-batch training, especially.It is equivalent between $\mathcal{E} = \sum_b^{\text{batch}} \mathcal{E}_b$ and $Log\mathcal{E} \neq \sum_b^{\text{batch}} Log\mathcal{E}_b$ with dynamic coefficients, as shown below:

$$dLog\mathcal{E}_b = d\ln \mathcal{E}_b^{II} + d\ln \mathcal{E}_b^I = \frac{1}{\mathcal{E}_b^{II}} d\mathcal{E}_b^{II} + \frac{1}{\mathcal{E}_b^I} d\mathcal{E}_b^I$$

Take $N = 2$ as an example, the goal is to minimize

$$\min \mathcal{E}^I \ \& \ \min \alpha^{II} = \frac{\mathcal{E}^{II}}{\mathcal{E}^I} - 1$$

Notice minimize $\alpha^{II}$ equals to minimize $\frac{\mathcal{E}^{II}}{\mathcal{E}^I}$ which can be converted to log format as minimized $\log(\mathcal{E}^{II}) - log(\mathcal{E}^I)$. Remember we still need minimize $\mathcal{E}^I$ which is same as minimize $\log(\mathcal{E}^I)$. These two part are independent implying that we need to assign two coefficients to combine them like:

$$\text{final loss} = a * [\log(\mathcal{E}^{II}) - log(\mathcal{E}^I)] + b * \log(\mathcal{E}^I)$$

They are hyper-parameter and we just set $a = 1, b = 2$ in this paper. This results in:

$$\text{final loss} = \log(\mathcal{E}^{II}) + \log(\mathcal{E}^I) = \text{MLSE}$$

Machine learning training typically involves batch updating. Thus, the key point here is to decide whether to "average after computing" or "compute after averaging". We leave this to the experiment.

For three-timestamps optimization MLSE, the derivation is:

$$\min \mathcal{E}^I \ \& \ \min \alpha^{II} = \frac{\mathcal{E}^{II}}{\mathcal{E}^I} - 1 \ \& \ \min \alpha^{III} = \frac{\mathcal{E}^{III}}{\mathcal{E}^{II}} - \frac{\mathcal{E}^I}{\mathcal{E}^{II}}$$

which is equivalent to:

$$\min\{\mathcal{E}^I, \frac{\mathcal{E}^{II}}{\mathcal{E}^I}, \frac{\mathcal{E}^{III}}{\mathcal{E}^{II}}\} \text{ and } \max\{\frac{\mathcal{E}^I}{\mathcal{E}^{II}}\}.$$

The last term is same as $\min\{\frac{\mathcal{E}^{II}}{\mathcal{E}^I}\}$. Using the hyper-parameter coefficient trick, we can finally obtain:

$$\text{MLSE}_{order3} = \log(\mathcal{E}^{III}) + \log(\mathcal{E}^{II}) + \log(\mathcal{E}^I)$$

In Equation 1, there are many other useful error information that can be utilized. For instance, we can calculate the error between $\boldsymbol{X}_{t+2}^{II}$ and $\boldsymbol{X}_{t+2}^I$ instead of $||\mathcal{E}_{t+2}^{II} = \boldsymbol{X}_{t+2}^O - \boldsymbol{X}_{t+2}^{II}||$. Notice the equation $\ln[\alpha_t^{II}] = \ln[||\boldsymbol{X}_{t+2}^{II} - \boldsymbol{X}_{t+2}^I||] - \ln[||\boldsymbol{X}_{t+1}^I - \boldsymbol{X}_{t+1}^O||] = \ln[\varepsilon_{t+2}^{II,I}] - \ln[\mathcal{E}_{t+1}^I]$ (as shown in Equation 2), thus a loss simultaneously optimizing $\ln \alpha_t^{II}$ and $\ln \mathcal{E}_{t+1}^I$ is equivalent to optimizing $\ln \varepsilon_{t+2}^{II,I}$ and $\ln \mathcal{E}_{t+1}^I$.

$$\begin{pmatrix} \boldsymbol{X}_t^O & \to \boldsymbol{X}_{t+1}^I & \to \boldsymbol{X}_{t+2}^{II} & \to \boldsymbol{X}_{t+3}^{III} & \to \boldsymbol{X}_{t+4}^{IV} \\ \boldsymbol{X}_{t+1}^O & \to \boldsymbol{X}_{t+2}^I & & & \\ \boldsymbol{X}_{t+2}^O & \to \boldsymbol{X}_{t+3}^I & & & \\ \boldsymbol{X}_{t+3}^O & \to \boldsymbol{X}_{t+4}^I & & & \end{pmatrix} \tag{3}$$

The ideas presented here have inspired us to develop the MASE

$$\text{MASE} = \sum_N \sum_t [||\boldsymbol{X}_{t+N}^N - \boldsymbol{X}_{t+N}^I|| + ||\boldsymbol{X}_{t+N-1}^{N\text{-}1} - \boldsymbol{X}_{t+N-1}^O||]$$

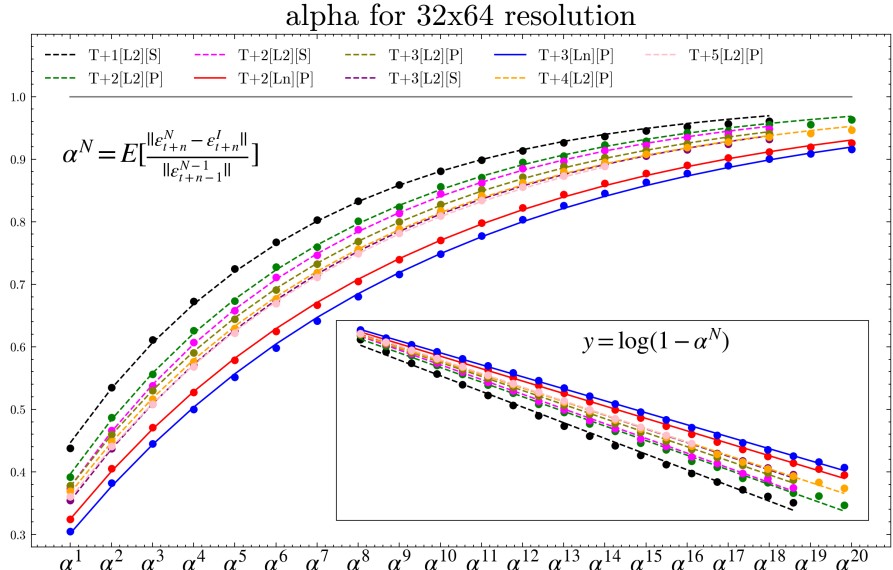

Figure 1: The figure shows the amplifier computed using Equation 2 for different convergence points resulting from various training strategies. The [S] label indicates that the model was trained from a random initialization, while the [P] label indicates that the model was trained from pre-trained weights of model T+1[L2][L2]. The [L2] label indicates that the mean squared error (MSE) loss was used, while the [Ln] label indicates that the mean logarithmic squared error (MLSE) loss was used. The T+n notation indicates that the error for the attention task (ATT) was optimized from 0 to N-th orders.

## 2  OBSERVATION

We monitor the amplifier $\alpha^N$ on the WeatherBench32x64 dataset for a well-trained FourCastNet model. We employ different training strategies to encourage the model to converge to different points. The test dataset for this evaluation consists of a $1480 \times (70 \times 32 \times 64)$ vector sequence. For each starting point $\boldsymbol{X}_t$, we calculate the model's forecast results up to 19 time steps, i.e., $\hat{X}t+1, \cdots, \hat{X}t+19$. Finally, we collect $1461 \times 18$ different order error $\mathcal{E}^N$ and calculate the amplifier $\alpha^N$ by

$$\alpha^N = E[\frac{\varepsilon_{t+N}^N - \varepsilon_{t+N}^I}{\varepsilon_{t+N-1}^{N-1}}] = E[\frac{||\boldsymbol{X}_{t+N}^N - \boldsymbol{X}_{t+N}^I||}{||\boldsymbol{X}_{t+N-1}^{N-1} - \boldsymbol{X}_{t+N-1}^O||}] \tag{4}$$

The results, depicted in Fig.1, are based on many different training strategies: The [S] label indicates that the model was trained from a random initialization, while the [P] label indicates that the model was trained from pre-trained weights of model T+1[L2][L2]. The [L2] label indicates that the mean squared error (MSE) loss was used, while the [Ln] label indicates that the mean logarithmic squared error (MLSE) loss was used. The T+n notation indicates that the error for the attention task (ATT) was optimized from 0 to N-th orders.

The findings depicted in Fig. 1 provide clear evidence that, once a deep learning model has been trained and converged on a dataset, its long-range prediction errors follow an exponential amplifier law. Specifically, the amplifier $\alpha^N$ between $N$-order errors and $N-1$ errors can be approximated by a geometric fitting curve, such as $\alpha^N = 1 - \beta_1 \exp(-\beta_2 * N)$. We validate this law for different models using varying training strategies, highlighting the fact that the long-term prediction error can be effectively optimized by

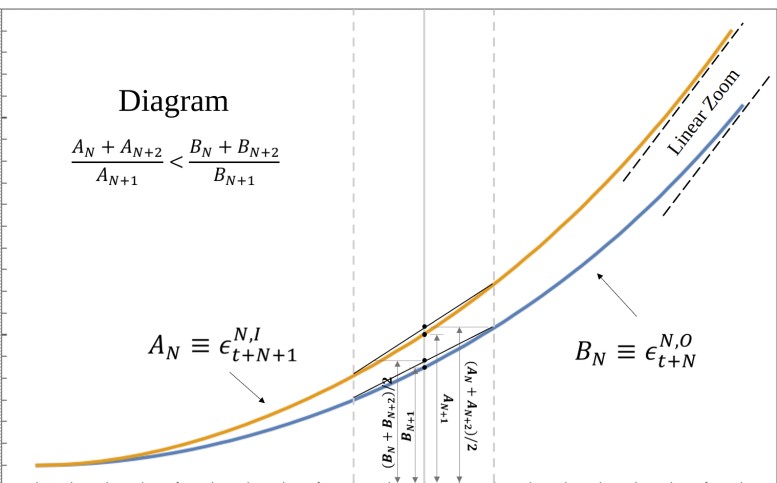

Figure 2: The diagram to show the convex of error propagation. Note it is only a diagram, to vivid demonstrate the convex of amplifier $\alpha^N = \frac{A_N}{B_N}$.

appropriately configuring the short-term error. These findings provide important insights into the relationship between short-term performance and long-term forecasting accuracy, particularly in cases where the model is not exposed to the forecasted data during training.

The geometric formulation exhibits an exponential behavior experimentally. However, a simple analysis reveals that the convex behavior can be obtained. Let us denote the series $||\varepsilon_{t+N+1}^{N+1,I}||$ as $A_N$ and the series $||\varepsilon_{t+N}^{N,O}||$ as $B_N$. We know that both $A_N$ and $B_N$ increase and achieve linear growth at infinity $N$; Before that, they are strictly convex. Since $A_N$ is the higher error incresement, we can assume $B_{N+2} \geq A_N \geq B_{N+1}$ for any $N$. This leads to $\frac{A_{N+2}+A_N}{A_{N+1}} \leq \frac{B_{N+2}+B_N}{B_{N+1}}$, as the error accumulation speed should be approximately consistent for both series $A_N$ and $B_N$ as shown in Diagram.2. Therefore, the second differences of the sequence $\alpha^N = \frac{A_N}{B_N}$ becomes

$$(\alpha^{n+2} - \alpha^{n+1}) - (\alpha^{n+1} - \alpha^n) = \frac{A_{n+2}B_{n+1} - A_{n+1}B_{n+2} - A_{n+1}B_n + A_nB_{n+1}}{B_nB_{n+1}B_{n+2}} \leq 0$$

. As the model is a black-box, it is difficult to obtain further insights into its behavior beyond the geometric formulation. However, we are able to analyze the asymptotic behavior as $N \to \infty$. Due to the fact that time-correlation is not infinite, the prediction at $N-1$ step $\boldsymbol{X}_{t+N}^N$ is no longer related to the first step $\boldsymbol{X}_{t+N-1}^O$, as well as $\boldsymbol{X}_{t+N}^N$ to $\boldsymbol{X}_{t+N}^I$. Consequently, if we assume each normlized element $X_{t,s}^N \sim N(0,1)$ follow the normal distribution, then the $\alpha^N$ degenerates to the division between two $\chi_k^2$ distributions, which corresponds to an $F$-distribution. The expectation is $\alpha^{N\to\infty} = \frac{S}{S-2} \to 1$, where $S$ is the freedom of $X$. This finding implies that the asymptotic behavior of error propagation is linear, which means the difference between $\varepsilon_N$ and $\varepsilon_{N-1}$ is a constant as $N \to \infty$. By combining this with $\alpha^{N\to\infty} = 1$, we can qualitatively understand the behavior of alpha.

The geometric behavior can aid in quickly estimating the long-term performance of a model based on a few first-order errors. For example, Fig. 3 shows the estimated performance of model [T+3][P][L2]. Due to the entanglement problem mentioned previously, $\alpha_t^N$ and $||\varepsilon_{t+N-1}^{N-1}||$ typically have the relationship $E[\alpha_t^N]E[||\varepsilon_{t+N-1}^{N-1}||] \leq E[\alpha_t^N]E[||\varepsilon_{t+N-1}^{N-1}||]$, which represents an upper bound on the true error. Fig. 3

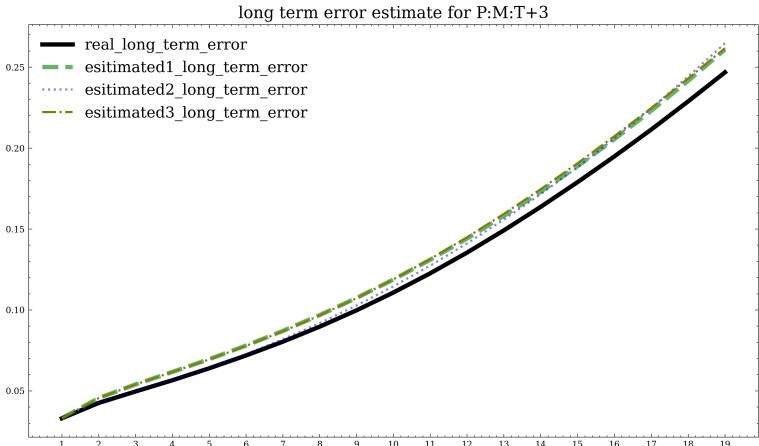

Figure 3: The estimated long-term performance of model [T+3][P][L2] from its first three low order errros. The three esimated method is low-order expansion, full-amplifier computing, and statistic-fitting. The result demonstrates that estimation can achieve a considerable approximation of the true long-term error by only measuring a few short-term errors. In this example, the 19-step error is smaller than 6%.

depicts the results obtained from three different estimation methods that were used to test the influence of different statistical routes: (1) The estimated1 method uses the statistical means $\alpha^0$ and $\alpha^1$ by only accessing the $\mathcal{E}^{\mathrm{I}}$, $\mathcal{E}^{\mathrm{II}}$, and $\mathcal{E}^{\mathrm{III}}$, then calculates all $\alpha$ values, and finally accumulates the amplifier using the Equ.2. (2) The estimated2 method uses all statistical means $\alpha^{n:1\to19}$. This is not a practical method for estimation since it requires the computation of all $\mathcal{E}^{n:1\to19}$. However, we plot the results to emphasize the feasibility of the exponential law for $\alpha$. (3) The estimated3 method uses the losses $\mathcal{E}^{\mathrm{I}}$, $\mathcal{E}^{\mathrm{II}}$, and $\mathcal{E}^{\mathrm{III}}$ directly to compute the "mean statistical alpha" $\bar{\alpha}^1 = \frac{\mathcal{E}^{\mathrm{II}}-\mathcal{E}^{\mathrm{I}}}{\mathcal{E}^{\mathrm{I}}}$ and $\bar{\alpha}^2 = \frac{\mathcal{E}^{\mathrm{III}}-\mathcal{E}^{\mathrm{I}}}{\mathcal{E}^{\mathrm{II}}}$. The example of model [T+3][P][L2] demonstrates that estimation can achieve a considerable approximation of the true long-term error by only measuring a few short-term errors. In this example, the 19-step error is smaller than 6%.

In this section, we have shown that the error amplifier in autoregressive prediction tasks follows an obvious geometric law. This implies that constraints on one amplifier will spontaneously distribute to all amplifiers. Therefore, the correct descent direction for minimizing long-term prediction error is the same as the direction for minimizing the amplifier directly. When the model $\theta$ is near the convergence point, it may be better to constrain the amplifier $\alpha$ and the 1-order error $\mathcal{E}^{\mathrm{I}}$ rather than directly constraining high-order errors $\mathcal{E}^{\mathrm{I}}, \mathcal{E}^{\mathrm{II}}, \mathcal{E}^{\mathrm{III}}$. We have also demonstrated that this phenomenological theory can help estimate long-range forecast performance by only measuring a few short-term errors. Unfortunately, even if we can analytically write the relationship between long-term and short-term errors, we cannot use this analytical expression for descent updates directly for several reasons: (1) It is merely an estimate and is only effective near the convergence point. (2) It is only effective in a statistical sense and is incompatible with the mini-batch training mode.(3) The gradient of the analytical expression for $\mathcal{E}^{\mathrm{II}}$ is always negative.

# 3 THE ALPHA MONITOR FOR TIME SERIES DATASET

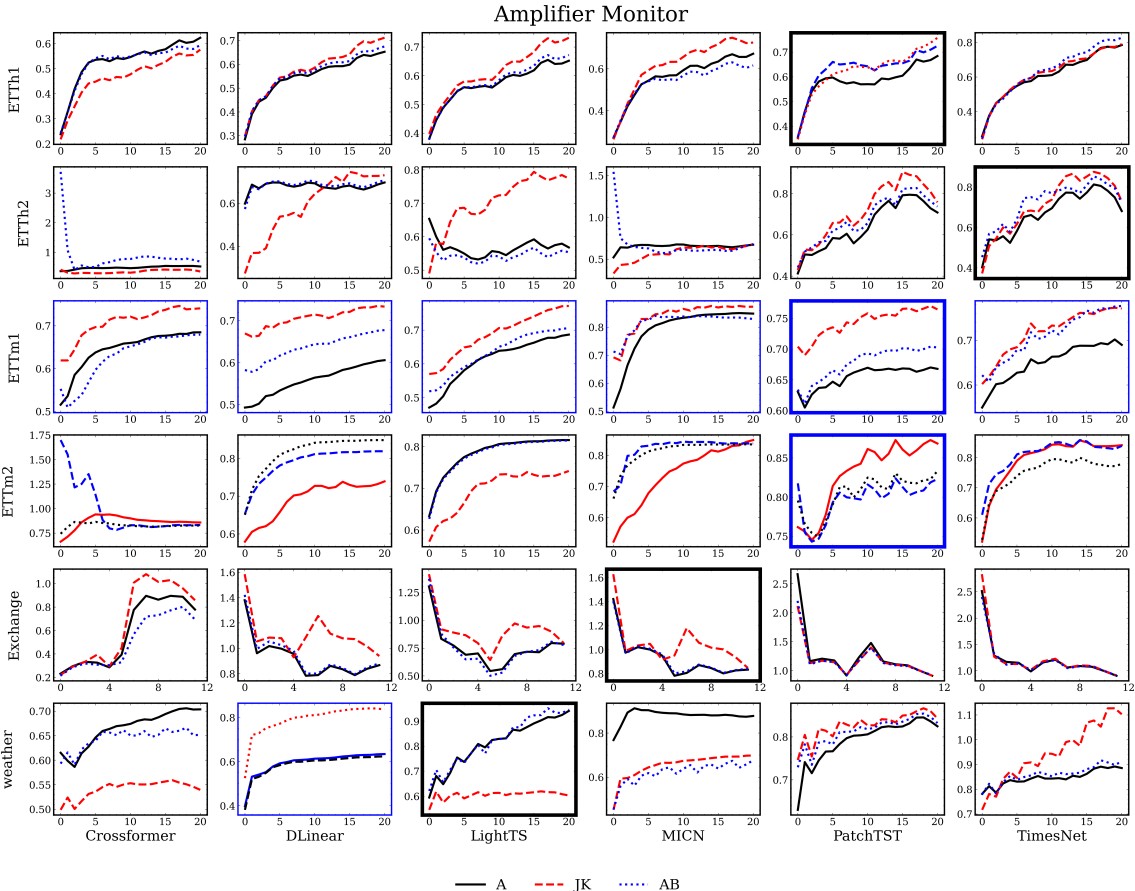

Figure 4: The alpha monitor for the case presented in Table 2 follows the same procedure outlined in Section 2. We only plot the alpha on the prediction sequence equal to 96. The frame colour and size of the 720 rows in Table 2 correspond. A black board indicates that the MLSE+ATT model performs better than the end-to-end model, while a blue board is the opposite. The bold boundary represents the best model for each dataset, which is the case with background shading in Table 2. From this figure, we can see that, except for the Exchange dataset, all alpha relations are approximately monotonically increasing and converge near 1. There are several abnormal examples, such as Crossformer under ETTh2 and ETTm2 datasets. By observing its long-term error, we can see that the end-to-end performance of this model is extremely poor and does not converge at all. However, using MLSE can correct this behavior and bring the long-term prediction into the same convergence basin. The exchange dataset performs differently from other time series datasets, implying that it is not a predictable system since economic systems are often filled with complex external factors.

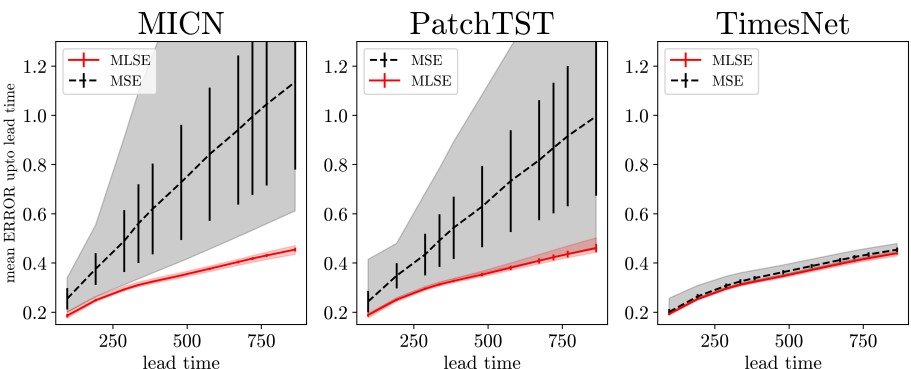

Figure 5: We evaluated the fine-tuned performance of TimesNet, MICN, and PatchTST on the ETTm2 datasets using MSE/MLSE metrics. The learning rate was adjusted from 1e-5 to 1e-3, and the random seed was selected from the following options: [19940928, 19950929, 20130901, 20230901]. On the y-axis, we plotted the mean error up to the lead time, which is the average error across the entire sequence. The shaded background represents the min-max region of error. The error bars denote the standard deviation, and the central line indicates the mean value. We conducted 20 samples for both the MLE and MLSE hyperparameter searches.

| dataset | pred | Crossformer A | D | R | DLinear A | D | R | LightTS A | D | R | MICN A | D | R | PatchTST A | D | R | TimesNet A | D | R |
|---|---|---|---|---|---|---|---|---|---|---|---|---|---|---|---|---|---|---|---|
| ETTh1 | 096 | 0.414 | 0.403 | 0.372 | 0.396 | 0.365 | 0.365 | 0.435 | 0.399 | 0.400 | 0.394 | 0.407 | 0.372 | 0.378 | 0.348 | 0.349 | 0.389 | 0.363 | 0.364 |
| | 192 | 0.450 | 0.480 | 0.426 | 0.445 | 0.414 | 0.412 | 0.494 | 0.441 | 0.438 | 0.454 | 0.474 | 0.434 | 0.437 | 0.382 | 0.389 | 0.437 | 0.396 | 0.403 |
| | 336 | 0.731 | 0.554 | 0.474 | 0.490 | 0.443 | 0.434 | 0.552 | 0.469 | 0.456 | 0.598 | 0.565 | 0.493 | 0.466 | 0.401 | 0.403 | 0.490 | 0.413 | 0.427 |
| | 720 | 0.609 | 0.736 | 0.613 | 0.509 | 0.525 | 0.485 | 0.613 | 0.561 | 0.520 | 0.696 | 0.794 | 0.693 | 0.507 | 0.456 | 0.426 | 0.520 | 0.461 | 0.489 |
| ETTh2 | 096 | 0.617 | 1.046 | 0.504 | 0.348 | 0.257 | 0.211 | 0.428 | 0.360 | 0.265 | 0.344 | 0.344 | 0.211 | 0.306 | 0.218 | 0.209 | 0.321 | 0.261 | 0.235 |
| | 192 | 1.585 | 4.579 | 0.817 | 0.479 | 0.361 | 0.254 | 0.582 | 0.510 | 0.311 | 0.494 | 0.545 | 0.252 | 0.377 | 0.266 | 0.250 | 0.390 | 0.300 | 0.261 |
| | 336 | 2.680 | 5.426 | 1.120 | 0.596 | 0.523 | 0.297 | 0.688 | 0.714 | 0.383 | 0.593 | 0.790 | 0.296 | 0.415 | 0.320 | 0.297 | 0.437 | 0.333 | 0.296 |
| | 720 | 3.497 | 4.293 | 1.549 | 0.825 | 1.007 | 0.368 | 1.006 | 1.164 | 0.595 | 0.827 | 1.342 | 0.370 | 0.449 | 0.406 | 0.364 | 0.487 | 0.391 | 0.356 |
| ETTm1 | 096 | 0.374 | 0.446 | 0.347 | 0.345 | 0.327 | 0.334 | 0.402 | 0.374 | 0.371 | 0.324 | 0.316 | 0.305 | 0.323 | 0.306 | 0.309 | 0.336 | 0.371 | 0.334 |
| | 192 | 0.448 | 0.636 | 0.433 | 0.382 | 0.367 | 0.369 | 0.431 | 0.417 | 0.414 | 0.363 | 0.376 | 0.353 | 0.369 | 0.347 | 0.348 | 0.385 | 0.408 | 0.372 |
| | 336 | 0.715 | 0.874 | 0.548 | 0.414 | 0.423 | 0.410 | 0.466 | 0.472 | 0.467 | 0.409 | 0.455 | 0.413 | 0.399 | 0.398 | 0.389 | 0.415 | 0.466 | 0.415 |
| | 720 | 0.687 | 1.273 | 0.822 | 0.472 | 0.541 | 0.488 | 0.561 | 0.584 | 0.570 | 0.509 | 0.654 | 0.584 | 0.459 | 0.496 | 0.471 | 0.476 | 0.563 | 0.521 |
| ETTm2 | 096 | 0.284 | 0.324 | 0.212 | 0.190 | 0.197 | 0.184 | 0.208 | 0.211 | 0.197 | 0.190 | 0.206 | 0.176 | 0.178 | 0.192 | 0.184 | 0.189 | 0.199 | 0.191 |
| | 192 | 0.374 | 0.905 | 0.300 | 0.275 | 0.289 | 0.251 | 0.317 | 0.301 | 0.261 | 0.274 | 0.298 | 0.243 | 0.241 | 0.262 | 0.252 | 0.251 | 0.262 | 0.251 |
| | 336 | 0.803 | 1.954 | 0.418 | 0.367 | 0.411 | 0.316 | 0.383 | 0.425 | 0.324 | 0.364 | 0.409 | 0.308 | 0.321 | 0.324 | 0.316 | 0.328 | 0.323 | 0.310 |
| | 720 | 3.200 | 5.902 | 0.709 | 0.550 | 0.724 | 0.419 | 0.778 | 0.742 | 0.424 | 0.541 | 0.681 | 0.415 | 0.401 | 0.426 | 0.418 | 0.422 | 0.421 | 0.408 |
| Exchange | 096 | 0.288 | 0.507 | 0.389 | 0.094 | 0.094 | 0.110 | 0.141 | 0.206 | 0.098 | 0.093 | 0.097 | 0.091 | 0.084 | 0.079 | 0.080 | 0.102 | 0.122 | 0.106 |
| | 192 | 0.661 | 0.785 | 0.487 | 0.185 | 0.189 | 0.179 | 0.310 | 0.553 | 0.187 | 0.185 | 0.191 | 0.151 | 0.184 | 0.147 | 0.147 | 0.211 | 0.236 | 0.200 |
| | 336 | 1.288 | 1.102 | 0.573 | 0.342 | 0.338 | 0.251 | 0.478 | 0.995 | 0.307 | 0.337 | 0.336 | 0.209 | 0.347 | 0.220 | 0.218 | 0.346 | 0.357 | 0.291 |
| | 720 | 1.708 | 1.268 | 0.738 | 0.747 | 0.602 | 0.638 | 0.937 | 1.231 | 0.576 | 0.802 | 0.609 | 0.482 | 0.847 | 0.632 | 0.632 | 0.903 | 0.850 | 0.767 |
| weather | 096 | 0.175 | 0.171 | 0.169 | 0.196 | 0.191 | 0.199 | 0.173 | 0.168 | 0.168 | 0.185 | 0.183 | 0.189 | 0.175 | 0.171 | 0.176 | 0.169 | 0.168 | 0.164 |
| | 192 | 0.232 | 0.213 | 0.207 | 0.235 | 0.225 | 0.237 | 0.234 | 0.206 | 0.201 | 0.254 | 0.215 | 0.222 | 0.222 | 0.213 | 0.219 | 0.228 | 0.210 | 0.207 |
| | 336 | 0.277 | 0.266 | 0.247 | 0.282 | 0.268 | 0.275 | 0.266 | 0.254 | 0.257 | 0.278 | 0.252 | 0.257 | 0.279 | 0.260 | 0.268 | 0.283 | 0.257 | 0.255 |
| | 720 | 0.368 | 0.363 | 0.310 | 0.344 | 0.359 | 0.354 | 0.344 | 0.356 | 0.297 | 0.351 | 0.333 | 0.324 | 0.355 | 0.346 | 0.358 | 0.354 | 0.343 | 0.366 |

Figure 6: A table similar to Table 2, but with the learning rate initialized at 2e-5 (original value is 1e-5).

| dataset | pred | Crossformer A | D | R | DLinear A | D | R | LightTS A | D | R | MICN A | D | R | PatchTST A | D | R | TimesNet A | D | R |
|---|---|---|---|---|---|---|---|---|---|---|---|---|---|---|---|---|---|---|---|
| ETTh1 | 096 | 0.414 | 0.384 | 0.363 | 0.396 | 0.386 | 0.366 | 0.435 | 0.402 | 0.402 | 0.394 | 0.389 | 0.372 | 0.378 | 0.348 | 0.349 | 0.389 | 0.361 | 0.362 |
| | 192 | 0.450 | 0.444 | 0.411 | 0.445 | 0.416 | 0.413 | 0.494 | 0.443 | 0.439 | 0.454 | 0.447 | 0.431 | 0.437 | 0.382 | 0.387 | 0.437 | 0.396 | 0.400 |
| | 336 | 0.731 | 0.504 | 0.451 | 0.490 | 0.446 | 0.437 | 0.552 | 0.470 | 0.457 | 0.598 | 0.523 | 0.490 | 0.466 | 0.401 | 0.401 | 0.490 | 0.415 | 0.424 |
| | 720 | 0.609 | 0.676 | 0.570 | 0.509 | 0.529 | 0.495 | 0.613 | 0.560 | 0.523 | 0.696 | 0.736 | 0.696 | 0.507 | 0.455 | 0.427 | 0.520 | 0.464 | 0.487 |
| ETTh2 | 096 | 0.617 | 1.031 | 0.562 | 0.348 | 0.260 | 0.215 | 0.428 | 0.359 | 0.266 | 0.344 | 0.304 | 0.212 | 0.306 | 0.213 | 0.207 | 0.321 | 0.255 | 0.234 |
| | 192 | 1.585 | 3.068 | 0.863 | 0.479 | 0.366 | 0.257 | 0.582 | 0.505 | 0.312 | 0.494 | 0.534 | 0.253 | 0.377 | 0.262 | 0.248 | 0.390 | 0.291 | 0.261 |
| | 336 | 2.680 | 4.664 | 1.135 | 0.596 | 0.533 | 0.299 | 0.688 | 0.709 | 0.384 | 0.593 | 0.799 | 0.295 | 0.415 | 0.317 | 0.296 | 0.437 | 0.327 | 0.296 |
| | 720 | 3.497 | 4.357 | 1.529 | 0.825 | 1.026 | 0.370 | 1.006 | 1.163 | 0.592 | 0.827 | 1.269 | 0.377 | 0.449 | 0.405 | 0.365 | 0.487 | 0.391 | 0.358 |
| ETTm1 | 096 | 0.374 | 0.398 | 0.348 | 0.345 | 0.327 | 0.332 | 0.402 | 0.377 | 0.376 | 0.324 | 0.315 | 0.305 | 0.323 | 0.306 | 0.307 | 0.336 | 0.352 | 0.332 |
| | 192 | 0.448 | 0.505 | 0.425 | 0.382 | 0.370 | 0.371 | 0.431 | 0.421 | 0.421 | 0.363 | 0.373 | 0.353 | 0.369 | 0.348 | 0.345 | 0.385 | 0.389 | 0.368 |
| | 336 | 0.715 | 0.639 | 0.524 | 0.414 | 0.428 | 0.420 | 0.466 | 0.476 | 0.475 | 0.409 | 0.453 | 0.415 | 0.399 | 0.398 | 0.387 | 0.415 | 0.444 | 0.409 |
| | 720 | 0.687 | 0.908 | 0.775 | 0.472 | 0.547 | 0.515 | 0.561 | 0.586 | 0.575 | 0.509 | 0.678 | 0.592 | 0.459 | 0.496 | 0.469 | 0.476 | 0.545 | 0.500 |
| ETTm2 | 096 | 0.284 | 0.310 | 0.215 | 0.190 | 0.200 | 0.184 | 0.208 | 0.211 | 0.197 | 0.190 | 0.203 | 0.176 | 0.178 | 0.189 | 0.183 | 0.189 | 0.196 | 0.192 |
| | 192 | 0.374 | 0.702 | 0.296 | 0.275 | 0.296 | 0.252 | 0.317 | 0.299 | 0.262 | 0.274 | 0.263 | 0.244 | 0.241 | 0.258 | 0.249 | 0.251 | 0.262 | 0.252 |
| | 336 | 0.803 | 1.466 | 0.407 | 0.367 | 0.425 | 0.318 | 0.383 | 0.417 | 0.325 | 0.364 | 0.335 | 0.308 | 0.321 | 0.321 | 0.312 | 0.328 | 0.320 | 0.310 |
| | 720 | 3.200 | 4.152 | 0.743 | 0.550 | 0.755 | 0.421 | 0.778 | 0.719 | 0.425 | 0.541 | 0.533 | 0.417 | 0.401 | 0.423 | 0.411 | 0.422 | 0.420 | 0.408 |
| Exchange | 096 | 0.288 | 0.476 | 0.387 | 0.094 | 0.095 | 0.114 | 0.141 | 0.196 | nan | 0.093 | 0.097 | 0.091 | 0.084 | 0.079 | 0.081 | 0.102 | 0.122 | 0.106 |
| | 192 | 0.661 | 0.685 | 0.481 | 0.185 | 0.189 | 0.186 | 0.310 | 0.510 | nan | 0.185 | 0.190 | 0.153 | 0.184 | 0.147 | 0.148 | 0.211 | 0.217 | 0.201 |
| | 336 | 1.288 | 0.922 | 0.563 | 0.342 | 0.337 | 0.263 | 0.478 | 0.924 | nan | 0.337 | 0.331 | 0.214 | 0.347 | 0.221 | 0.220 | 0.346 | 0.318 | 0.293 |
| | 720 | 1.708 | 1.118 | 0.745 | 0.747 | 0.599 | 0.708 | 0.937 | 1.190 | nan | 0.802 | 0.599 | 0.484 | 0.847 | 0.642 | 0.639 | 0.903 | 0.779 | 0.774 |
| weather | 096 | 0.175 | 0.172 | 0.168 | 0.196 | 0.191 | 0.197 | 0.173 | 0.168 | 0.167 | 0.185 | 0.184 | 0.187 | 0.175 | 0.172 | 0.174 | 0.169 | 0.168 | 0.165 |
| | 192 | 0.232 | 0.216 | 0.206 | 0.235 | 0.225 | 0.234 | 0.234 | 0.205 | 0.200 | 0.254 | 0.217 | 0.219 | 0.222 | 0.213 | 0.218 | 0.228 | 0.210 | 0.207 |
| | 336 | 0.277 | 0.268 | 0.246 | 0.282 | 0.268 | 0.275 | 0.266 | 0.250 | 0.237 | 0.278 | 0.255 | 0.253 | 0.279 | 0.260 | 0.266 | 0.283 | 0.256 | 0.255 |
| | 720 | 0.368 | 0.368 | 0.308 | 0.344 | 0.359 | 0.350 | 0.344 | 0.341 | 0.296 | 0.351 | 0.331 | 0.317 | 0.355 | 0.345 | 0.354 | 0.354 | 0.340 | 0.367 |

Figure 7: A table similar to Table 2, but with the random seed initialized at 2023. (original value is 2021)

| | | ETTh1 | | | | ETTh2 | | | | ETTm1 | | | | ETTm2 | | | | Exchange | | | | Weather | | | |
|---|---|---|---|---|---|---|---|---|---|---|---|---|---|---|---|---|---|---|---|---|---|---|---|---|---|
| | | 096 | 192 | 336 | 720 | 096 | 192 | 336 | 720 | 096 | 192 | 336 | 720 | 096 | 192 | 336 | 720 | 096 | 192 | 336 | 720 | 096 | 192 | 336 | 720 |
| Autoformer | A | 0.467 | 0.533 | 0.535 | 0.503 | 0.360 | 0.443 | 0.478 | 0.482 | 0.480 | 0.614 | 0.539 | 0.719 | 0.272 | 0.292 | 0.337 | 0.422 | 0.159 | 0.287 | 0.471 | 1.117 | 0.264 | 0.302 | 0.360 | 0.418 |
| | D | 0.416 | 0.437 | 0.450 | 0.489 | 0.257 | 0.301 | 0.338 | 0.427 | 0.443 | 0.482 | 0.552 | 0.762 | 0.241 | 0.307 | 0.395 | 0.654 | 0.160 | 0.249 | 0.372 | 1.119 | 0.209 | 0.279 | 0.381 | 0.725 |
| | R | 0.413 | 0.464 | 0.496 | 0.586 | 0.232 | 0.271 | 0.313 | 0.385 | 0.425 | 0.481 | 0.551 | 0.764 | 0.223 | 0.279 | 0.332 | 0.425 | 0.159 | 0.244 | 0.335 | 0.986 | 0.209 | 0.272 | 0.341 | 0.552 |
| Crossformer | A | 0.414 | 0.450 | 0.731 | 0.609 | 0.617 | 1.585 | 2.680 | 3.497 | 0.374 | 0.448 | 0.715 | 0.687 | 0.284 | 0.374 | 0.803 | 3.200 | 0.288 | 0.661 | 1.288 | 1.708 | 0.175 | 0.232 | 0.277 | 0.368 |
| | D | 0.367 | 0.418 | 0.468 | 0.631 | 0.586 | 0.915 | 1.334 | 2.017 | 0.364 | 0.444 | 0.541 | 0.710 | 0.296 | 0.500 | 0.785 | 1.570 | 0.461 | 0.635 | 0.803 | 1.029 | 0.172 | 0.215 | 0.266 | 0.363 |
| | R | 0.363 | 0.410 | 0.444 | 0.552 | 0.533 | 0.830 | 1.110 | 1.491 | 0.340 | 0.415 | 0.506 | 0.728 | 0.216 | 0.298 | 0.414 | 0.774 | 0.391 | 0.488 | 0.572 | 0.730 | 0.165 | 0.204 | 0.243 | 0.305 |
| DLinear | A | 0.396 | 0.445 | 0.490 | 0.509 | 0.348 | 0.479 | 0.596 | 0.825 | 0.345 | 0.382 | 0.414 | 0.472 | 0.190 | 0.275 | 0.367 | 0.550 | 0.094 | 0.185 | 0.342 | 0.747 | 0.196 | 0.235 | 0.282 | 0.344 |
| | D | 0.366 | 0.416 | 0.446 | 0.529 | 0.258 | 0.362 | 0.526 | 1.014 | 0.327 | 0.370 | 0.428 | 0.547 | 0.194 | 0.280 | 0.389 | 0.660 | 0.095 | 0.190 | 0.337 | 0.592 | 0.188 | 0.225 | 0.268 | 0.358 |
| | R | 0.366 | 0.413 | 0.437 | 0.496 | 0.215 | 0.257 | 0.300 | 0.371 | 0.331 | 0.371 | 0.420 | 0.515 | 0.184 | 0.252 | 0.318 | 0.421 | 0.097 | 0.163 | 0.225 | 0.512 | 0.188 | 0.226 | 0.272 | 0.349 |
| Informer | A | 0.994 | 0.923 | 1.107 | 1.185 | 3.577 | 5.442 | 5.029 | 3.876 | 0.643 | 0.792 | 1.215 | 1.160 | 0.744 | 1.050 | 1.300 | 3.884 | 0.911 | 1.096 | 1.765 | 2.698 | 0.368 | 0.766 | 0.549 | 1.534 |
| | D | 0.737 | 0.826 | 0.895 | 1.058 | 1.591 | 1.578 | 1.510 | 1.659 | 0.692 | 0.722 | 0.786 | 0.838 | 0.816 | 0.858 | 0.910 | 1.042 | 1.130 | 1.000 | 0.917 | 0.937 | 0.279 | 0.330 | 0.328 | 0.340 |
| | R | 0.762 | 0.844 | 0.897 | 0.997 | 1.590 | 1.579 | 1.510 | 1.659 | 0.585 | 0.639 | 0.725 | 0.818 | 0.815 | 0.867 | 0.923 | 1.017 | 1.059 | 0.950 | 0.885 | 0.885 | 0.279 | 0.311 | 0.326 | 0.350 |
| LightTS | A | 0.435 | 0.494 | 0.552 | 0.613 | 0.428 | 0.582 | 0.688 | 1.006 | 0.402 | 0.431 | 0.466 | 0.561 | 0.208 | 0.317 | 0.383 | 0.778 | 0.141 | 0.310 | 0.478 | 0.937 | 0.173 | 0.234 | 0.266 | 0.344 |
| | D | 0.402 | 0.443 | 0.469 | 0.557 | 0.351 | 0.495 | 0.693 | 1.118 | 0.376 | 0.420 | 0.474 | 0.582 | 0.211 | 0.299 | 0.414 | 0.705 | 0.165 | 0.378 | 0.655 | 0.942 | 0.168 | 0.206 | 0.250 | 0.338 |
| | R | 0.401 | 0.439 | 0.457 | 0.521 | 0.265 | 0.311 | 0.382 | 0.586 | 0.376 | 0.421 | 0.475 | 0.576 | 0.197 | 0.262 | 0.325 | 0.425 | 0.103 | 0.196 | 0.320 | 0.594 | 0.167 | 0.200 | 0.237 | 0.296 |
| MICN | A | 0.394 | 0.454 | 0.598 | 0.696 | 0.338 | 0.494 | 0.593 | 0.827 | 0.322 | 0.358 | 0.393 | 0.506 | 0.190 | 0.270 | 0.355 | 0.535 | 0.092 | 0.184 | 0.327 | 0.795 | 0.183 | 0.239 | 0.275 | 0.349 |
| | D | 0.362 | 0.403 | 0.449 | 0.620 | 0.262 | 0.353 | 0.488 | 0.852 | 0.313 | 0.364 | 0.437 | 0.624 | 0.195 | 0.262 | 0.333 | 0.528 | 0.097 | 0.186 | 0.318 | 0.565 | 0.184 | 0.216 | 0.254 | 0.329 |
| | R | 0.362 | 0.403 | 0.448 | 0.620 | 0.212 | 0.252 | 0.293 | 0.369 | 0.306 | 0.354 | 0.415 | 0.588 | 0.176 | 0.243 | 0.306 | 0.409 | 0.091 | 0.153 | 0.214 | 0.453 | 0.184 | 0.217 | 0.250 | 0.311 |
| PatchTST | A | 0.378 | 0.437 | 0.466 | 0.507 | 0.306 | 0.377 | 0.415 | 0.449 | 0.323 | 0.369 | 0.399 | 0.459 | 0.178 | 0.241 | 0.321 | 0.401 | 0.084 | 0.184 | 0.347 | 0.847 | 0.175 | 0.222 | 0.279 | 0.355 |
| | D | 0.346 | 0.381 | 0.401 | 0.455 | 0.211 | 0.259 | 0.313 | 0.398 | 0.306 | 0.347 | 0.398 | 0.495 | 0.184 | 0.250 | 0.313 | 0.415 | 0.078 | 0.145 | 0.216 | 0.613 | 0.171 | 0.213 | 0.260 | 0.345 |
| | R | 0.349 | 0.386 | 0.400 | 0.426 | 0.206 | 0.248 | 0.296 | 0.365 | 0.306 | 0.345 | 0.387 | 0.468 | 0.182 | 0.248 | 0.311 | 0.409 | 0.079 | 0.145 | 0.215 | 0.606 | 0.175 | 0.217 | 0.264 | 0.350 |
| Pyraformer | A | 0.767 | 0.857 | 0.984 | 1.017 | 1.719 | 5.877 | 5.193 | 4.637 | 0.526 | 0.616 | 0.849 | 0.851 | 0.325 | 0.633 | 1.334 | 4.588 | 0.678 | 1.118 | 1.259 | 1.957 | 0.185 | 0.243 | 0.295 | 0.411 |
| | D | 0.758 | 0.837 | 0.891 | 0.996 | 1.450 | 2.304 | 2.697 | 3.101 | 0.567 | 0.619 | 0.678 | 0.806 | 0.353 | 0.678 | 1.248 | 2.440 | 0.977 | 1.190 | 1.144 | 1.111 | 0.187 | 0.235 | 0.296 | 0.412 |
| | R | 0.750 | 0.841 | 0.904 | 1.022 | 1.189 | 2.314 | 2.697 | 3.109 | 0.585 | 0.637 | 0.706 | 0.856 | 0.274 | 0.414 | 0.634 | 1.155 | 0.978 | 1.145 | 1.116 | 1.133 | 0.166 | 0.206 | 0.251 | 0.324 |
| Reformer | A | 0.815 | 0.969 | 0.974 | 1.216 | 1.698 | 2.586 | 2.756 | 3.019 | 0.738 | 0.887 | 1.006 | 1.081 | 0.859 | 1.699 | 1.619 | 3.257 | 1.054 | 1.500 | 1.977 | 2.128 | 0.365 | 0.784 | 0.559 | 0.942 |
| | D | 0.725 | 0.776 | 0.817 | 0.940 | 1.430 | 1.974 | 2.541 | 2.936 | 0.709 | 0.779 | 0.840 | 0.901 | 0.788 | 0.913 | 1.152 | 1.759 | 0.465 | 0.628 | 0.750 | 0.953 | 0.311 | 0.401 | 0.484 | 0.594 |
| | R | 0.726 | 0.777 | 0.817 | 0.940 | 1.428 | 1.943 | 2.164 | 2.544 | 0.709 | 0.779 | 0.841 | 0.901 | 0.788 | 0.913 | 1.152 | 1.756 | 0.531 | 0.672 | 0.774 | 0.974 | 0.304 | 0.392 | 0.468 | 0.562 |
| Stationary | A | 0.495 | 0.606 | 0.738 | 0.672 | 0.425 | 0.510 | 0.559 | 0.551 | 0.407 | 0.510 | 0.568 | 0.625 | 0.259 | 0.555 | 0.441 | 0.603 | 0.134 | 0.220 | 0.370 | 1.188 | 0.197 | 0.250 | 0.345 | 0.418 |
| | D | 0.463 | 0.496 | 0.539 | 0.627 | 0.306 | 0.334 | 0.369 | 0.421 | 0.389 | 0.454 | 0.531 | 0.667 | 0.253 | 0.323 | 0.390 | 0.512 | 0.120 | 0.219 | 0.321 | 0.847 | 0.191 | 0.238 | 0.289 | 0.403 |
| | R | 0.451 | 0.486 | 0.521 | 0.613 | 0.308 | 0.335 | 0.369 | 0.421 | 0.389 | 0.454 | 0.531 | 0.666 | 0.253 | 0.324 | 0.391 | 0.513 | 0.106 | 0.213 | 0.307 | 0.783 | 0.179 | 0.236 | 0.289 | 0.403 |
| TimesNet | A | 0.389 | 0.437 | 0.490 | 0.520 | 0.321 | 0.390 | 0.437 | 0.487 | 0.336 | 0.385 | 0.415 | 0.476 | 0.189 | 0.251 | 0.328 | 0.422 | 0.102 | 0.211 | 0.346 | 0.903 | 0.169 | 0.228 | 0.283 | 0.354 |
| | D | 0.360 | 0.395 | 0.414 | 0.462 | 0.234 | 0.271 | 0.320 | 0.388 | 0.332 | 0.374 | 0.428 | 0.528 | 0.191 | 0.254 | 0.315 | 0.414 | 0.104 | 0.184 | 0.265 | 0.706 | 0.166 | 0.208 | 0.254 | 0.340 |
| | R | 0.360 | 0.397 | 0.420 | 0.478 | 0.230 | 0.260 | 0.296 | 0.357 | 0.327 | 0.364 | 0.406 | 0.497 | 0.192 | 0.252 | 0.310 | 0.409 | 0.104 | 0.184 | 0.265 | 0.704 | 0.165 | 0.207 | 0.254 | 0.341 |
| Transformer | A | 0.839 | 0.865 | 1.011 | 0.944 | 3.185 | 5.279 | 5.441 | 3.394 | 0.609 | 0.735 | 1.086 | 1.183 | 0.526 | 1.016 | 1.371 | 2.917 | 0.549 | 0.959 | 1.450 | 1.924 | 0.328 | 0.519 | 0.702 | 0.933 |
| | D | 0.680 | 0.772 | 0.821 | 0.914 | 1.956 | 2.195 | 2.026 | 1.983 | 0.601 | 0.596 | 0.613 | 0.642 | 0.719 | 0.864 | 0.941 | 1.058 | 0.474 | 0.621 | 0.727 | 0.882 | 0.322 | 0.385 | 0.458 | 0.618 |
| | R | 0.697 | 0.756 | 0.790 | 0.867 | 2.236 | 2.190 | 2.070 | 2.000 | 0.584 | 0.595 | 0.613 | 0.642 | 0.530 | 0.651 | 0.720 | 0.853 | 0.438 | 0.590 | 0.677 | 0.775 | 0.305 | 0.381 | 0.447 | 0.594 |

Table 1: The training information of weatherbench in this paper.

## 5 NEGITIVE EXAMPLE IN TIME SERIES DATASET

The validation of MLSE is heavily depend the time property of the dataset. One example is the traffic dataset who has clear periodicity, The Table.8 shows that MLSE is generally ineffective, and most models tend to use end-to-end systems to handle long-term forecasting rather than ATT.

| model | | Autoformer | | | Crossformer | | | DLinear | | | Informer | | | LightTS | | | MICN | | | PatchTST | | | Pyraformer | | | Reformer | | | Stationary | | | TimesNet | | | Transformer | | |
|---|---|---|---|---|---|---|---|---|---|---|---|---|---|---|---|---|---|---|---|---|---|---|---|---|---|---|---|---|---|---|---|---|---|---|---|---|---|---|
| metric | | A | D | R | A | D | R | A | D | R | A | D | R | A | D | R | A | D | R | A | D | R | A | D | R | A | D | R | A | D | R | A | D | R | A | D | R |
| dataset | pred | | | | | | | | | | | | | | | | | | | | | | | | | | | | | | | | | | | | |
| traffic | 096 | 0.671 | 0.685 | 0.694 | 0.526 | 0.520 | 0.531 | 0.699 | 0.700 | 0.696 | 0.721 | 0.720 | 0.758 | 0.682 | 0.667 | 0.680 | 0.521 | 0.510 | 0.522 | 0.454 | 0.429 | 0.462 | 0.695 | 0.662 | 0.694 | 0.716 | 0.699 | 0.740 | 0.614 | 0.614 | 0.643 | 0.588 | 0.590 | 0.590 | 0.641 | 0.628 | 0.667 |
| | 192 | 0.664 | 0.666 | 0.679 | 0.565 | 0.536 | 0.546 | 0.647 | 0.683 | 0.683 | 0.736 | 0.716 | 0.765 | 0.671 | 0.662 | 0.702 | 0.536 | 0.531 | 0.549 | 0.461 | 0.435 | 0.491 | 0.685 | 0.666 | 0.711 | 0.701 | 0.685 | 0.728 | 0.647 | 0.618 | 0.648 | 0.617 | 0.598 | 0.598 | 0.662 | 0.635 | 0.666 |
| | 336 | 0.618 | 0.674 | 0.685 | 0.634 | 0.561 | 0.570 | 0.653 | 0.719 | 0.712 | 0.831 | 0.718 | 0.767 | 0.691 | 0.689 | 0.745 | 0.550 | 0.558 | 0.580 | 0.476 | 0.466 | 0.542 | 0.696 | 0.678 | 0.742 | 0.704 | 0.683 | 0.727 | 0.654 | 0.631 | 0.670 | 0.638 | 0.613 | 0.631 | 0.662 | 0.643 | 0.667 |
| | 720 | 0.658 | 0.707 | 0.718 | 0.592 | 0.601 | 0.611 | 0.694 | 0.804 | 0.786 | 0.957 | 0.741 | 0.788 | 0.744 | 0.784 | 0.873 | 0.576 | 0.600 | 0.623 | 0.509 | 0.532 | 0.635 | 0.721 | 0.711 | 0.778 | 0.700 | 0.689 | 0.735 | 0.660 | 0.680 | 0.749 | 0.659 | 0.645 | 0.694 | 0.696 | 0.662 | 0.678 |

Figure 8: The table shows the complete experimental results for end-to-end, ATT-MSE, and ATT-MLSE loss performance on the traffic dataset. Metric $A$ represents the baseline MSE performance of the end-to-end pre-trained model, while Metrics $D$ and $R$ represent the forecast performance after applying ATT training processing with MSE and MLSE as optimize loss, respectively.

# 6  THE ERROR BAR (STD) FOR TIME SERIES DATASET

We fix the random seed and repeat the finetune experiment 5 times to get the standard deviation of the results. (These fluctuations arise from machine and build-in code feature). The results are shown in Table 9.

| dataset | pred | Autoformer D | Autoformer R | Crossformer D | Crossformer R | DLinear D | DLinear R | Informer D | Informer R | LightTS D | LightTS R | MICN D | MICN R | PatchTST D | PatchTST R | Pyraformer D | Pyraformer R | Reformer D | Reformer R | Stationary D | Stationary R | TimesNet D | TimesNet R | Transformer D | Transformer R |
|---|---|---|---|---|---|---|---|---|---|---|---|---|---|---|---|---|---|---|---|---|---|---|---|---|---|
| ETTh1 | 096 | 2e-03 | 2e-03 | 3e-03 | 1e-03 | 1e-05 | 1e-05 | 4e-03 | 4e-03 | 2e-04 | 2e-04 | 1e-03 | 2e-03 | 3e-04 | 0e+00 | 4e-03 | 0e+00 | 7e-03 | 5e-03 | 4e-03 | 2e-03 | 5e-04 | 2e-05 | 6e-03 | 4e-03 |
|  | 192 | 3e-03 | 6e-03 | 5e-03 | 2e-03 | 2e-05 | 1e-05 | 3e-03 | 3e-03 | 6e-04 | 5e-04 | 3e-03 | 5e-03 | 3e-04 | 4e-04 | 1e-02 | 8e-03 | 8e-03 | 5e-03 | 2e-03 | 3e-03 | 3e-04 | 4e-04 | 1e-02 | 9e-03 |
|  | 336 | 6e-03 | 9e-03 | 9e-03 | 4e-03 | 5e-05 | 3e-05 | 5e-03 | 3e-03 | 1e-03 | 1e-03 | 6e-03 | 8e-03 | 3e-04 | 2e-04 | 2e-02 | 9e-03 | 7e-03 | 5e-03 | 3e-04 | 4e-03 | 6e-04 | 8e-04 | 1e-02 | 1e-02 |
|  | 720 | 6e-03 | 1e-02 | 1e-02 | 1e-02 | 2e-04 | 2e-04 | 1e-02 | 6e-03 | 3e-03 | 3e-03 | 1e-02 | 2e-02 | 2e-03 | 4e-04 | 2e-02 | 5e-03 | 2e-03 | 1e-03 | 1e-03 | 9e-03 | 1e-03 | 2e-03 | 2e-02 | 8e-03 |
| ETTh2 | 096 | 4e-04 | 2e-04 | 1e-02 | 7e-03 | 5e-04 | 2e-04 | 7e-01 | 5e-01 | 4e-03 | 5e-04 | 1e-03 | 4e-04 | 7e-04 | 6e-04 | 4e-02 | 2e-02 | 7e-02 | 6e-02 | 2e-03 | 3e-03 | 3e-05 | 4e-04 | 3e-02 | 7e-02 |
|  | 192 | 2e-03 | 4e-04 | 3e-03 | 1e-02 | 1e-03 | 2e-04 | 6e-01 | 4e-01 | 1e-02 | 8e-04 | 1e-02 | 1e-04 | 7e-04 | 5e-04 | 8e-01 | 5e-01 | 8e-03 | 1e-02 | 3e-03 | 2e-03 | 4e-05 | 6e-04 | 3e-02 | 5e-02 |
|  | 336 | 2e-02 | 7e-04 | 2e-02 | 1e-02 | 2e-03 | 2e-04 | 5e-01 | 4e-01 | 2e-02 | 2e-03 | 3e-02 | 3e-04 | 7e-04 | 6e-04 | 7e-01 | 5e-01 | 1e-01 | 1e-02 | 2e-03 | 2e-03 | 1e-03 | 6e-04 | 2e-02 | 4e-02 |
|  | 720 | 7e-02 | 4e-03 | 6e-02 | 1e-02 | 3e-03 | 1e-04 | 4e-01 | 3e-01 | 2e-02 | 4e-03 | 1e-02 | 1e-03 | 1e-03 | 4e-04 | 3e-01 | 1e-01 | 2e-02 | 2e-02 | 2e-03 | 2e-03 | 1e-03 | 6e-04 | 2e-02 | 3e-02 |
| ETTm1 | 096 | 4e-03 | 9e-04 | 5e-03 | 1e-03 | 6e-06 | 4e-05 | 2e-04 | 4e-03 | 4e-05 | 8e-05 | 4e-03 | 2e-04 | 5e-04 | 4e-04 | 2e-02 | 0e+00 | 2e-02 | 1e-02 | 6e-03 | 2e-03 | 4e-04 | 5e-04 | 4e-03 | 5e-03 |
|  | 192 | 1e-02 | 2e-03 | 6e-03 | 2e-03 | 2e-05 | 2e-05 | 2e-02 | 4e-03 | 2e-04 | 1e-04 | 1e-02 | 2e-03 | 4e-04 | 5e-04 | 2e-02 | 0e+00 | 2e-02 | 1e-02 | 9e-03 | 3e-03 | 3e-03 | 6e-04 | 6e-03 | 6e-03 |
|  | 336 | 2e-02 | 1e-02 | 1e-02 | 5e-03 | 4e-05 | 4e-05 | 2e-02 | 3e-03 | 6e-04 | 2e-04 | 2e-02 | 4e-04 | 1e-03 | 3e-04 | 1e-02 | 0e+00 | 2e-02 | 1e-02 | 1e-02 | 5e-03 | 6e-03 | 1e-03 | 7e-03 | 8e-03 |
|  | 720 | 5e-02 | 4e-02 | 3e-02 | 1e-02 | 8e-05 | 9e-05 | 4e-03 | 7e-03 | 1e-03 | 3e-04 | 6e-02 | 2e-03 | 2e-03 | 7e-04 | 2e-02 | 2e-03 | 2e-02 | 1e-02 | 2e-02 | 1e-02 | 2e-02 | 1e-02 | 1e-02 | 1e-02 |
| ETTm2 | 096 | 7e-03 | 4e-04 | 9e-04 | 9e-04 | 3e-17 | 3e-06 | 2e-02 | 1e-16 | 1e-03 | 4e-06 | 1e-07 | 7e-05 | 1e-03 | 2e-05 | 8e-03 | 1e-03 | 3e-02 | 3e-02 | 5e-03 | 4e-03 | 4e-04 | 2e-04 | 3e-03 | 2e-03 |
|  | 192 | 7e-03 | 5e-04 | 1e-02 | 2e-03 | 6e-17 | 1e-05 | 2e-02 | 0e+00 | 3e-03 | 1e-05 | 8e-03 | 1e-04 | 3e-04 | 5e-04 | 3e-02 | 5e-03 | 3e-02 | 3e-02 | 4e-03 | 4e-03 | 2e-04 | 3e-04 | 4e-03 | 3e-03 |
|  | 336 | 6e-03 | 7e-04 | 2e-02 | 3e-03 | 0e+00 | 2e-05 | 2e-02 | 1e-16 | 6e-03 | 3e-05 | 2e-02 | 5e-04 | 2e-04 | 7e-04 | 9e-02 | 1e-02 | 3e-02 | 3e-02 | 4e-03 | 4e-03 | 3e-04 | 4e-04 | 5e-03 | 4e-03 |
|  | 720 | 6e-04 | 1e-03 | 2e-02 | 1e-02 | 1e-16 | 3e-05 | 2e-02 | 8e-03 | 2e-02 | 4e-05 | 7e-02 | 1e-03 | 1e-03 | 6e-04 | 1e-01 | 2e-02 | 3e-02 | 3e-02 | 2e-02 | 2e-02 | 9e-04 | 3e-04 | 2e-03 | 5e-03 |
| Exchange | 096 | 4e-03 | 3e-03 | 2e-02 | 4e-03 | 1e-04 | 0e+00 | 2e-02 | 3e-02 | 0e+00 | 6e-04 | 5e-04 | 9e-04 | 2e-04 | 5e-04 | 3e-02 | 5e-02 | 2e-01 | 1e-01 | 7e-04 | 1e-03 | 3e-05 | 6e-04 | 6e-04 | 1e-02 |
|  | 192 | 8e-03 | 5e-03 | 4e-02 | 6e-03 | 9e-04 | 2e-03 | 9e-03 | 2e-02 | 0e+00 | 2e-03 | 5e-03 | 1e-03 | 7e-04 | 8e-04 | 9e-02 | 1e-01 | 2e-01 | 1e-01 | 1e-03 | 3e-03 | 2e-04 | 9e-05 | 5e-03 | 7e-03 |
|  | 336 | 2e-02 | 7e-03 | 5e-02 | 9e-03 | 2e-03 | 3e-03 | 9e-03 | 2e-02 | 0e+00 | 6e-03 | 4e-02 | 2e-03 | 9e-04 | 2e-03 | 7e-02 | 9e-02 | 2e-01 | 1e-01 | 2e-03 | 4e-03 | 3e-04 | 3e-04 | 5e-03 | 1e-02 |
|  | 720 | 5e-02 | 3e-02 | 4e-02 | 1e-02 | 2e-03 | 2e-02 | 9e-03 | 2e-02 | 1e-16 | 2e-03 | 2e-02 | 2e-02 | 9e-03 | 1e-02 | 2e-02 | 2e-02 | 2e-01 | 8e-02 | 2e-02 | 2e-02 | 6e-04 | 4e-05 | 5e-03 | 1e-02 |
| weather | 096 | 2e-02 | 8e-03 | 1e-03 | 3e-04 | 0e+00 | 0e+00 | 2e-02 | 1e-03 | 0e+00 | 2e-04 | 3e-03 | 5e-03 | 2e-03 | 2e-03 | 3e-03 | 4e-04 | 8e-03 | 2e-03 | 2e-03 | 1e-03 | 5e-05 | 3e-04 | 2e-03 | 8e-03 |
|  | 192 | 2e-02 | 6e-04 | 4e-04 | 3e-04 | 2e-04 | 3e-17 | 6e-03 | 1e-03 | 2e-04 | 4e-05 | 3e-03 | 2e-03 | 3e-04 | 1e-03 | 4e-03 | 2e-04 | 8e-03 | 3e-03 | 3e-03 | 9e-05 | 2e-04 | 6e-04 | 6e-04 | 1e-02 |
|  | 336 | 4e-02 | 3e-03 | 8e-04 | 3e-04 | 4e-04 | 2e-04 | 7e-03 | 2e-03 | 6e-04 | 8e-05 | 6e-03 | 1e-03 | 6e-05 | 7e-04 | 6e-03 | 7e-04 | 6e-03 | 3e-03 | 5e-03 | 4e-03 | 1e-04 | 1e-02 | 6e-03 | 1e-02 |
|  | 720 | 5e-02 | 2e-02 | 1e-04 | 4e-04 | 9e-04 | 6e-04 | 7e-03 | 3e-03 | 5e-03 | 3e-04 | 7e-03 | 5e-03 | 1e-04 | 5e-05 | 1e-02 | 4e-03 | 3e-03 | 4e-03 | 2e-02 | 2e-02 | 4e-04 | 3e-04 | 2e-02 | 2e-02 |

Figure 9: The table shows the standard derivation for repeated experience among all the time series experiment. We only repeat experiment for finetune task.

# 7 USE MLSE START FROM SKETCH DOESN'T PERFORM BETTER

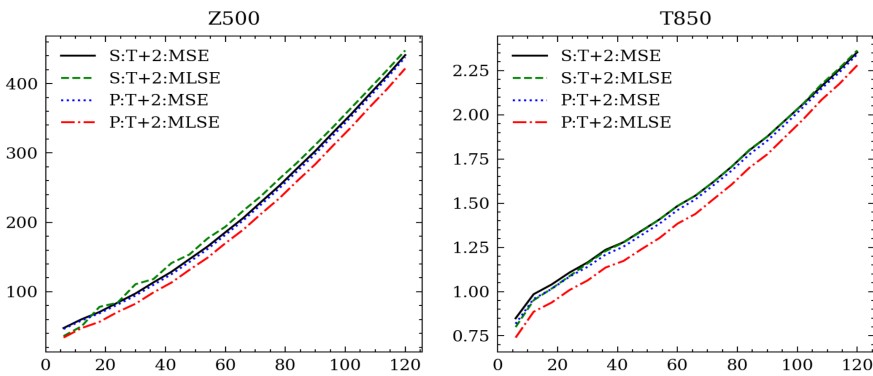

Figure 10: Directly use MLSE and train a model from sketch has no any benefit. It is only a finetune technology.

# 8 EULER EQUATION

The latent inductive bias in physical simulation tasks is that all evolution must follow a physics formula. For weather simulation, this can be an explicit formula such as the Euler equation(Equ.8)Laprise (1992) or am unexplict neural network simulator such as FourCast. This prior implies that the relation between time frames must be "Space to Time", meaning that next state variables such as velocity $V_{t+1}$, geometry $\phi_{t+1}$ and temperature $T_{t+1}$ can be derived from the current state information $V_t$, $\phi_t$ and $T_t$.

$$\partial_t V = F - V \cdot \nabla V - \omega \partial_p V - \nabla \phi$$

$$\partial_t T = Q/C_v + \frac{RT}{C_p}\omega - V \cdot \nabla T - \omega \partial_p T$$

$$\partial_t \phi = wg - V \cdot \nabla V - \omega \partial_p \phi$$

where $V$ is the horizontal velocity, $\omega$ is the vertical velocity driven by $\frac{\partial \omega}{\partial p} = -\nabla \cdot V$, $\phi$ is the geopotential, $T$ is the temperature, $F$ is the force, all of which are continuous functions of the cartesian coordinates $(x, y)$ and pressure $p$. Additionally, $Q$ is the external heat, $C_v$ is the heat capacity and $R$ is the gas constant.

## 9 DATASET INFORMATION

In this paper, we have two major dataset:

• **WeatherBench**Rasp et al. (2020) is a large, high-resolution global atmospheric dataset created to facilitate the development and benchmarking of machine learning models for weather forecasting. It is widely used in the AI for weather predicting community, such as PanguBi et al. (2022), GraphCastLam et al. (2022) and FengwuChen et al. (2023). The dataset includes a range of atmospheric variables, such as temperature ($T$), pressure ($\phi$), humidity ($H$) and wind speed ($V$) at multiple levels of the atmosphere, and covers the entire globe with a resolution of approximately 0.25 degrees (or about 28 km). The data is available at hourly intervals and covers a period from 1979 to 2018. This paper only involves the $32x64$ and $64x128$ resolutions. We use the horizontal wind speed information, temperature, pressure and humidity data, but without the constant variables; these features consist of the $68$ channels of one earth snapshot. Following the same habbit as previous worksBi et al. (2022); Lam et al. (2022); Chen et al. (2023), we sequentially divide the train/valid/test datasets. More specifically, we take data from years 1979-2016 as the train dataset, year 2017 as the valid dataset, and year 2018 as the test dataset. The WeatherBench-300k dataset task hourly snapshot thus has around three hundred thousands rawdata. The WeatherBench-50k dataset downsamples hourly data into 6-hourly data and holds around 55k rawdata.

| dataset | WeatherBench32x64-6hour | | | WeatherBench32x64-1hour | | | WeatherBench64x128-1hour | | |
|---|---|---|---|---|---|---|---|---|---|
| shape | $68 \times 32 \times 64$ | | | $68 \times 32 \times 64$ | | | $68 \times 32 \times 128$ | | |
| split | train | valid | test | train | valid | test | train | valid | test |
| years | 1979 - 2016 | 2017 | 2018 | 1979 - 2015 | 2016 - 2017 | 2018 | 1979 - 2015 | 2016 - 2017 | 2018 |
| number | 55514 | 1458 | 1458 | 324301 | 17533 | 8749 | 324301 | 17533 | 8749 |

Table 2: The information of weatherbench dataset used in this paper.

• **Time series** use the long-term forecasting setting in TimesNet Wu et al. (2022) and same hyperparameters for benchmark. The datasets consist of ETT Zhou et al. (2021), Electricity Trindade (2016), Traffic PeM, Weather Wet, and Exchange Lai et al. (2018). Each part is a single continuous time series, and we sample rawdata by a sliding window.

| dataset | input feature | train | valid | test | categorize |
|---|---|---|---|---|---|
| ETTm1, ETTm2 | $7 \times 96, 192, 336, 720$ | 34465 | 11521 | 11521 | Electricity(15 mins) |
| ETTh1, ETTh2 | $7 \times 96, 192, 336, 720$ | 8545 | 2881 | 2881 | Electricity (15 mins) |
| Electricity | $321 \times 96, 192, 336, 720$ | 18317 | 2633 | 5261 | Electricity (Hourly) |
| Traffic | $862 \times 96, 192, 336, 720$ | 12185 | 1757 | 3509 | Transportation (Hourly) |
| Weather | $21 \times 96, 192, 336, 720$ | 36792 | 5271 | 10540 | Weather (10 mins) |
| Exchange | $8 \times 96, 192, 336, 720$ | 5120 | 665 | 1422 | Exchange rate (Daily) |

Table 3: The information of time series dataset used in this paper.

## 10 HYPER-PARAMETER AND TRAINING DETAIL

For the WeatherBench dataset training, the same hyperparameters are used for the settings "MSE", "MASE" or "MLSE". We usually train the model on 4 or 8 A100-80G GPUs using a DataParallel pipeline. The batch size is calculated as the total number and the random seed is fixed at 73001. We use a cosine learning rate scheduler which will anneal the learning rate from 1e-6 to the set learning rate in the warmup epoch and then fall back to 1e-5. The optimizer is AdamW with parameters $\beta = (0.9, 0.95)$ and weight decay 0.05. More details can be found in Table 10. Due to computing resource limitations, we only train the model 64x128 with a small number of epochs. It can be seen that the model quickly overfits in the $64 \times 128$ resolution during the finetune procedure. We do not use an earlystop strategy, so all models run for the

| | dataset | WB32x64-50k | | WB32x64-300k | | | | WB64x128-300k | |
|---|---|---|---|---|---|---|---|---|---|
| | model | AFNONet | | AFNONet | | Lgnet | | Lgnet | |
| | phase | pre-T | fine-T | pre-T | fine-T | pre-T | fine-T | pre-T | fine-T |
| | epoch | 100 | 100 | 40 | 40 | 20 | 20 | 20 | 4 |
| | batch_size | 64 | 64 | 64 | 16 | 64 | 16 | 64 | 16 |
| optimz | type | AdamW | | | | | | | |
| | beta | (0.9, 0.95) | | | | | | | |
| sched | type | Cosine | | | | | | | |
| | warmup | 5 | | | | | | | |
| | value | 1e-6 → lr → 1e-5 | | | | | | | |
| | lr | 8e-4 | 8e-4 | 8e-4 | 8e-4 | 2e-4 | 2e-4 | 8e-4 | 1e-6 |

Table 4: The training information of weatherbench in this paper.

For the timeseries dataset, the hyperparameters for both the "MSE" and "MLSE" settings are identical. The only difference between the pretrain and finetune phases is the learning rate; the finetune learning rate is set to 1e-5, which is 10x smaller than the pretrain learning rate of 1e-4. All architecture settings are aligned with the TimesNet Library Wu et al. (2022). The scheduler is cosine with 0 warmup, so the learning rate will decay to 1e-6 from its set value. The batch size is listed in Table 10. All experiments were run on a single A100-80G GPU.

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

| dataset | ETTh1 | ETTh2 | ETTm1 | ETTm2 | Exchange | traffic | weather |
|---|---|---|---|---|---|---|---|
| Autoformer | 32 | 32 | 32 | 32 | 32 | 32 | 32 |
| Crossformer | 32 | 32 | 32 | 32 | 32 | 16 | 32 |
| DLinear | 32 | 32 | 32 | 32 | 32 | 32 | 32 |
| Informer | 32 | 32 | 32 | 32 | 32 | 32 | 32 |
| LightTS | 32 | 32 | 32 | 32 | 32 | 32 | 32 |
| MICN | 32 | 32 | 32 | 32 | 32 | 32 | 32 |
| PatchTST | 32 | 32 | 32 | 32 | 32 | 4 | 32 |
| Pyraformer | 32 | 32 | 32 | 32 | 32 | 32 | 32 |
| Reformer | 32 | 32 | 32 | 32 | 32 | 32 | 32 |
| Stationary | 32 | 32 | 32 | 32 | 32 | 32 | 32 |
| TimesNet | 32 | 32 | 32 | 32 | 32 | 32 | 32 |
| Transformer | 32 | 32 | 32 | 32 | 32 | 32 | 32 |

Table 5: The batch size of time series in this paper.

Guokun Lai, Wei-Cheng Chang, Yiming Yang, and Hanxiao Liu. Modeling long-and short-term temporal patterns with deep neural networks. In *The 41st international ACM SIGIR conference on research & development in information retrieval*, pp. 95–104, 2018.

Remi Lam, Alvaro Sanchez-Gonzalez, Matthew Willson, Peter Wirnsberger, Meire Fortunato, Alexander Pritzel, Suman Ravuri, Timo Ewalds, Ferran Alet, Zach Eaton-Rosen, et al. Graphcast: Learning skillful medium-range global weather forecasting. *arXiv preprint arXiv:2212.12794*, 2022.

René Laprise. The euler equations of motion with hydrostatic pressure as an independent variable. *Monthly weather review*, 120(1):197–207, 1992.

Jaideep Pathak, Shashank Subramanian, Peter Harrington, Sanjeev Raja, Ashesh Chattopadhyay, Morteza Mardani, Thorsten Kurth, David Hall, Zongyi Li, Kamyar Azizzadenesheli, et al. Fourcastnet: A global data-driven high-resolution weather model using adaptive fourier neural operators. *arXiv preprint arXiv:2202.11214*, 2022.

Stephan Rasp, Peter D Dueben, Sebastian Scher, Jonathan A Weyn, Soukayna Mouatadid, and Nils Thuerey. Weatherbench: a benchmark data set for data-driven weather forecasting. *Journal of Advances in Modeling Earth Systems*, 12(11):e2020MS002203, 2020.

Artur Trindade. Uci maching learning repository-electricityloaddiagrams20112014 data set, 2016.

Haixu Wu, Tengge Hu, Yong Liu, Hang Zhou, Jianmin Wang, and Mingsheng Long. Timesnet: Temporal 2d-variation modeling for general time series analysis. *arXiv preprint arXiv:2210.02186*, 2022.

Haoyi Zhou, Shanghang Zhang, Jieqi Peng, Shuai Zhang, Jianxin Li, Hui Xiong, and Wancai Zhang. Informer: Beyond efficient transformer for long sequence time-series forecasting. In *Proceedings of the AAAI conference on artificial intelligence*, volume 35, pp. 11106–11115, 2021.