# OpenReview forum: "The Logarithm Trick: achieve better long term forecast via Mean Logarithm Square Loss"
_ICLR.cc/2024/Conference — Submitted to ICLR 2024_

### Official Review · Reviewer_cW5f · 2023-10-26

**Soundness:** 2 fair
**Presentation:** 1 poor
**Contribution:** 2 fair
**Rating:** 3
**Confidence:** 2

**Summary:**

Time series prediction can be modeled as an autoregressive task and optimized through a pretraining-finetuning strategy. This paper proposes MLSE, an element-wise logarithmic operation following the standard square error loss, which enhances long-term forecast performance in the fine-tuning phase. MLSE acts as a plug-and-play enhancement for any autoregressive task.

**Strengths:**

1. The authors propose a simple yet effective method to enhance performance in long-term time series forecasting.
2. The proposed method performs well across a variety of datasets, forecasting models, and forecasting time steps.
3. The authors offer theoretical support for their proposed method.

**Weaknesses:**

1. Given that this paper focuses on the study of loss functions for time series forecasting, there should be comparisons with more time series loss functions. For example, see [1].
2. The quality of writing needs significant improvement. The paper contains numerous typos, grammatical mistakes, and incorrect LaTeX function usages. The equations, in particular, are difficult to follow. Refer to the questions below.
3. To have a deeper understanding of the proposed loss function, it would be beneficial to include traditional, straightforward models into the experiments, such as autoregression (AR), ARIMA, or DeepAR [2].

[1] A. Jadon. “A Comprehensive Survey of Regression Based Loss Functions for Time Series Forecasting.” arXiv 2022

[2] D. Salinas, “DeepAR: Probabilistic Forecasting with Autoregressive Recurrent Networks.” arXiv 2017.

**Questions:**

1. On page 4, what do T850 and Z500 represent?
2. What is the difference between the experiments done in Sections 3.1 and 3.2? Why have the authors chosen different forecasting models for each section?
3. There should be more explanation about the penultimate equation on page 6, which is about the propagation matrix M.

---

> ### Author Response · Authors · 2023-11-11
>
> # Thank you for your feedback.
> The MLSE loss function is specifically designed to address long-term forecasting problems by utilizing the concept of an 'error amplifier'. It is not a universal loss function for arbitrary timeseries tasks. We value your suggestions and are planning to extend our research to cover these interesting perspectives.
>
> ### Answer to question
> - We report the Root Mean Square Error (RMSE) in terms of two weather states, the T850 and the Z500, which are the crucial measurements in weather simulation tasks.
>
> - We aim to test the effectiveness of MLSE across different models, various dataset scales, and disparate dataset resolutions.
>
> - The $M_{t+N-1}^{N-1}$ is the propagation matrix produced by the Jacobian operator on multi-dimension Mean Value Theorem:.You can simply regard it as the $\nabla f_\theta({X}_{\delta})$. It is a matrix, if the input $X$ is a vector.

---

### Official Review · Reviewer_th6B · 2023-10-30

**Soundness:** 3 good
**Presentation:** 1 poor
**Contribution:** 2 fair
**Rating:** 5
**Confidence:** 3

**Summary:**

Authors introduce a new loss that is able to mitigate error-propagation in autoregressive systems which is an especially relevant problem for the time-series and spatio-temporal forecasting literatures. Their proposal is to use the logarithm of the Mean Squared Error (MSE) instead of the MSE alone which seems to help in time-series forecasting for some architectures and is competitive for spatio-temporal modeling (e.g. weather forecasting in this case). Moreover, the authors present a theoretical justification for their approach, which is based off the "error amplifier" that they try to minimize and using the theoretical justification, they arrive at different losses, of which, they chose the logarithm one for its superior empirical results.

**Strengths:**

- Authors present a plug-and-play method that requires a small change in the training loss, which makes its adoption as well as iterating over and comparing against much easier
- Extensive and rigourous experiments were done, both for weather forecasting which is a multi-dimensional problem and plain time-series forecasting. MLSE shows strong performance for some time-series architectures while it's competitive for weather forecasting.
- A theoretical justification for the MLSE is provided.
- MLSE is "ablated" against other variants from the theory section.

**Weaknesses:**

- Paper is poorly written and would benefit from a much needed overhaul:
  - \citep{} should be used instead of \citet{} whenever parenthetical citations are needed.
  - When referenceing figures, it's better to say "Figure 4" instead of "Figure4" or "Fig.4" which makes it more readable (same goes for tabkes as well).
  - In page 7, $\alpha^20$ should be replaced by $\alpha^{20}$.
  - Overall style could be improved.
- Although a theoretical justification is presented in section 4, some assumptions are made without justification:
   - While the orthogonality conjecture is true in high dimensions for random uniform vectors (https://math.stackexchange.com/questions/3059747/probability-of-two-random-points-being-orthogonal-in-higher-dimensional-unit-sph), it would be at least worth it to provide a justification for it in the appendix.
   - In equation 2, I don't understand how: $\frac{||\epsilon^N_{t+N} - \epsilon^I_{t+N}||}{||\epsilon^{N-1}_{t+N-1}||} \approx \frac{||\epsilon^N_{t+N}|| - ||\epsilon^I_{t+N}||}{||\epsilon^{N-1}_{t+N-1}||}$. No justification is provided for why the approximation holds. This seems to be very important for the theoretical derivations and conclusions made after.
- LMSE is the same as LMAE (up to a constant), so it would be fairer to include training under MAE loss as well.

**Questions:**

- Inconsistent error definition between section 2 and section 4 which can be confusing.
- For how many orders does the Dalpha strategy optimize for ? Does it pick one order ?
- How is MASE an improved alternative over Dalpha? They seem two different things altogether. Dalpha optimizes the relative errors while MASE optimizes each error separately
- I don't understand how the conclusion towards using the logarithm is made, in the paper it's just mentioned that "it should be", but that's not sufficient.

---

> ### Author Response · Authors · 2023-11-11
>
> -----------------
> Thank you for your feedback.
> $$
> \frac{||\epsilon_{t+N}^{N} - \epsilon_{t+N}^{I} ||}{||\epsilon_{t+N-1}^{N-1}||}\approx\frac{||\epsilon_{t+N}^{N}|| - || \epsilon_{t+N}^{I} ||}{||\epsilon_{t+N-1}^{N-1}||}
> $$
>
> due to the $|| {\epsilon}_{t+N}^{I} ||$ is the first order error and should be optimized in the pretrain phase for a pretrain-finetune diagram.
>
> and the $||\epsilon_{t+N}^{N} ||$ is yet optmized, shoud much large than $|| {\epsilon}_{t+N}^{I} ||$
>
> Questions:
>
> >  For how many orders does the Dalpha strategy optimize for ? Does it pick one order ?
>
> Figure 5 should the notation of Dalpha for first order and first + second order. Compute the Dalpha requires one-more forward operation (achieve t+N+1) compare to tradition long-term error (who only forward to t+N) and use the extra prediction compute the error. Thus, Dalpha-1 actually use the second order prediction (notice, not the error). And the Dalpha-2 use the third order prediction.
>
> > How is MASE an improved alternative over Dalpha? They seem two different things altogether. Dalpha optimizes the relative errors while MASE optimizes each error separately
>
> Lets take the first order as example, the full Dalpha requires optimize the e1 and e2'/e1. It requires spontaneously minimum the e1 and maximum e1.  This make training hard. In MASE we just remove the denominator.
>
> > I don't understand how the conclusion towards using the logarithm is made, in the paper it's just mentioned that "it should be", but that's not sufficient.
>
> In our paper, we approached the optimization problem by analyzing an amplifier due to the denominator. Therefore, we switched to optimizing its logarithmic form. Let's delve deeper into the derivation for **two timestamps optimization**:
> $$
> \min \\{\mathcal{E}^{I} \quad and \quad \alpha^{II}=\frac{\mathcal{E}^{II}}{\mathcal{E}^{I}}-1\\}
> $$
>   Notice minimize $\alpha^{II}$ equals to minimize $\frac{\mathcal{E}^{II}}{\mathcal{E}^{I}}$  which can be converted to log format as minimized  $\log(\mathcal{E}^{II}) - log(\mathcal{E}^{I})$. Remember we still need minimize $\mathcal{E}^{I}$ which is same as minimize $\log(\mathcal{E}^{I})$. These two part are independent implying that we need to assign two coefficients to combine them like:
> $$
> \text{final loss}=a*[\log(\mathcal{E}^{II}) - log(\mathcal{E}^{I})]+b*\log(\mathcal{E}^{I})
> $$
>   They are hyper-parameter and we just set the $a = 1, b = 2$ in this paper. This results in:
> $$
> \text{final loss}=\log(\mathcal{E}^{II}) + \log(\mathcal{E}^{I}) ={MLSE}
> $$
>
> It's crucial to note there is no explicit theoretical formulation mandating the use of the logarithm. The MASE can also be the feasibly accommodation for using amplifier information. However, it need an additional forward computation for an extra peek into the future. And we want a zero-cost operation. Therefore we name this technology as a `trick` and leave its validation in a comprehensive experiments.

---

> > ### Comment · Reviewer_th6B · 2023-11-21
> >
> > I still don't understand why the approximation holds in Equation 2. Approximations should always be justified with the right set of assumptions which are lacking in this case. Given that your method is based off it, it's only natural to rigourously justify it.
> >
> > My other points were addressed although my last point about comparing the MAE still holds, and it would be good to have additional experiments for it.

---

> > > ### Author Response · Authors · 2023-11-22
> > >
> > > As far as the MAE function, we quickly make three extra experiments, the result below
> > > |  time_step   | T->T+2 |        |        | T->T+3 |        |        |
> > > | :----------: | :----: | :----: | :----: | :----: | :----: | :----: |
> > > |   loss_fn    |   L1   |   Ln   |  MAE   |   L1   |   Ln   |  MAE   |
> > > | 6H(01step)   | 46.315 | 33.953 | 35.531 | 49.954 | 34.873 | 37.126 |
> > > | 12H(02step)  | 58.529 | 47.613 | 48.234 | 60.805 | 47.730 | 49.233 |
> > > | Day1(04step) | 81.847 | 71.277 | 69.627 | 81.749 | 70.288 | 70.215 |
> > > | Day3(12step) | 224.78 | 213.42 | 207.88 | 220.16 | 209.3  | 205.83 |
> > > | Day5(20step) | 436.9  | 421.95 | 414.41 | 424.26 | 412.11 | 408.82 |
> > >
> > > TL;DR:  The MLSE and MAE can both significantly outperform MSE. During the short-term stages, MLSE performs slightly better than MAE. Conversely, during the long-term stages, MAE tends to have a slight edge over MLSE.

---

> ### Author Response · Authors · 2023-11-21
>
> The equation $||\epsilon_{t+N}^{N} - \epsilon_{t+N}^{I} ||$ represents the difference between the error of a multistep forward prediction and the first step error. In the pretraining phase, we optimize the first step error $e_1=f(x)-y$ over a long period of epochs, resulting in a small error ($e_1$).
>
> In contrast, during the pretraining phase, we place no constraints on the multistep forward error like $e_N=f(...f(f(x)))) - y$, so at the beginning of the fine-tuning phase, it is larger ($e_N >> e_1$) because it has been under-optimized. You can consider it as a summation of large vector $\epsilon_{t+N}^{N} $  plus a perturbation vector $\epsilon_{t+N}^{I}$. And resulting
>  $||\epsilon_{t+N}^{N} - \epsilon_{t+N}^{I} || \approx  ||\epsilon_{t+N}^{N}|| - || \epsilon_{t+N}^{I} ||$.
>
> After the fine-tuning phase, this approximation remains valid for large $N$. For smaller $N$ (like $N=2$), the ratio $||\epsilon_{t+N}^{II}||/||\epsilon_{t+N}^{I}||$ is approximately $1.4$, indicating that the multistep forward error is larger, but not dramatically so.
>
> Indeed, we can also consider the approximation from the perspective of bounds. The term $||\epsilon_{t+N}^{N} - \epsilon_{t+N}^{I} ||$ can be bounded as:
>
> $$
> ||\epsilon_{t+N}^{N}|| - || \epsilon_{t+N}^{I} || \leq ||\epsilon_{t+N}^{N} - \epsilon_{t+N}^{I} || \leq ||\epsilon_{t+N}^{N}|| + || \epsilon_{t+N}^{I} ||
> $$
>
> This implies that the difference between the multistep forward error and the first step error lies between the multistep forward error minus the first step error and their sum. This offers another perspective to understand the formulation and can serve as the basis for developing a lower/upper bound approximation. We can then aim to optimize the model performance with respect to these bounds.

---

### Official Review · Reviewer_LBo8 · 2023-10-31

**Soundness:** 2 fair
**Presentation:** 3 good
**Contribution:** 2 fair
**Rating:** 5
**Confidence:** 4

**Summary:**

This paper focuses on the autoregressive prediction task. The key contribution is the proposed elementwise logarithmic operation following the standard square error loss, i.e., Mean Logarithm Square Error (MLSE), for improved forecasting performance. Generally extensive experiments are performed to support this claim.

**Strengths:**

1.	The proposed elementwise logarithmic operation following the standard square error loss, i.e., Mean Logarithm Square Error (MLSE), for improved long-term forecasting performance. Some theoretical analysis is provided.
2.	Generally extensive empirical studies.

**Weaknesses:**

1.	Although this paper proposes a simple and effective trick for autoregressive prediction, its significance is not enough for a ICLR publication.
2.	Experiments: In this paper the weather forecasting task is considered as a typical autoregressive prediction task. In fact, in weather forecasting we are interested in large n. For example, in the hourly forecasting, if we are interested in the weather forecast in the next 3 days, then we have n=72. From Table 1, when n increases, the performance advantage of the propose trick seems decrease for T+3. Thus, it would be very interesting to explore the comparison for some large values of n.

**Questions:**

N/A

---

> ### Author Response · Authors · 2023-11-11
>
> Thank you for your feedback.
> ---
> - We would like to emphasize that, to the best of our knowledge, this is the first paper that attempts to analyze the problem of error accumulation and proposes an effective and cost-free method to mitigate it.
> ---
> - The experimental setup in this paper sets one step, $n=1$, to be equivalent to 6 real-time hours. Therefore, $n=12$ spans over a period of 3 days and $n=20$ over 5 days.
> -  It is to be expected that the performance advantage of the proposed method decreases for T+3, T+4, and so forth. This occurs due to the rapid decay of time correlation between T and T+4, making the addition of more timestamp information considerably inefficient. Owing to resource constraints, we may not be able to make a comprehensive comparison for larger $n$ values.
>
> Please see follow is the `Z500` score for the `WeatherBench32x64-55k` dataset and `FourCastNet` model.
> The finetune phase start from same pretrain weight and run in same hyperparameter like lr and optimizer and 100 epochs.
> |   time_step  | T->T+2 |        | T->T+3 |        | T->T+4 |        | T->T+5 |        | T->T+6 |        |
> |:------------:|:------:|:------:|:------:|:------:|:------:|:------:|:------:|:------:|:------:|:------:|
> |    loss_fn   |   L1   |   Ln   |   L1   |   Ln   |   L1   |   Ln   |   L1   |   Ln   |   L1   |   Ln   |
> | Day1(04step) | 81.847 | 71.277 | 81.749 | 70.288 | 84.513 | 70.483 | 87.512 | 72.336 | 89.553 | 73.868 |
> | Day3(12step) | 224.78 | 213.42 | 220.16 |  209.3 | 223.16 | 207.47 | 227.28 | 209.56 | 228.04 | 213.89 |
> | Day5(20step) |  436.9 | 421.95 | 424.26 | 412.11 | 429.49 | 410.41 | 430.91 | 408.02 | 431.56 | 414.79 |
>
> You can see
> - The MLSE always outperform the MSE case.
> - The effect for more future information (T->T+n) is decay
>    - Notice, we directly add higher order term for one finetune phase in above experiment. We set this just for comparison. In practice, it is better do multiple finetune phase like first train for T->T+1, then T-> T+2 then T->T+3, and so on.

---

> ### Comment · Reviewer_LBo8 · 2023-11-23
> **Thanks for extra experiment and clarification.**
>
> Thanks for extra experiment and clarification. The experimental results also shows the good performance given large forecasting horizon. Overall, after reading the other reviews, I decide to increase my score but tend not to accept the paper.

---

### Official Review · Reviewer_YAoo · 2023-11-10

**Soundness:** 2 fair
**Presentation:** 2 fair
**Contribution:** 2 fair
**Rating:** 3
**Confidence:** 4

**Summary:**

This paper considers a particular approach for auto-regressive time series forecasting specifically, in which a model is pre-trained with next-step prediction and then fine-tuned for multiple auto-regressive prediction steps (for which predictions and errors are propagated).

The authors propose to use a mean log squared error (MLSE) loss for fine-tuning, rather than the basic mean squared error (MSE) loss - that is for updating the model further based on the error for more than just the next time step prediction (which is still in this setting less than the target horizon for which the models are evaluated).

The authors show with experiments on weather forecasting datasets and a couple non-weather time series datasets, for multiple different target horizons and base models, that using this MLSE loss consistently (for the majority of cases) offers some improvement (reduction in test error) compared to MSE fine-tuning.

They provide some analysis and justification for using this loss base on a concept of error amplification relating different-order errors (i.e., errors after multiple auto-regressive steps), and further discuss limitations.

**Strengths:**

-The experiments involved multiple datasets and horizons, and showed consistent improvement using MSLE vs MSE, which is interesting

-The problem and setting were well-motivated

-The proposed method is very practical, in that it requires only a very simple modification to the existing common loss function and approach used (i.e., simply taking the log of the squared errors)

**Weaknesses:**

- The novelty seems somewhat limited.  The only new thing introduced is adding a "log" transform in the fine tuning loss function - everything else is prior work. Mean squared log error is not a new metric or loss, and has been used for both training and evaluating forecast models before.  One could argue the connection to error accumulation in this particular setting (and the application particularly to fine-tuning) is the novel part, but this the justification for this does not seem so clear, aside from experiment results.


- The paper is not very clear or detailed.  E.g., the actually loss functions used in each training phase are not well defined, there are many details left out, such as how hyper parameters are selected and models tuned.  There are many incomplete sentences.  The details in the explanations / formulation are lacking and not clear.  E.g., many detail are left out in section 4 making it hard to follow.  It's not clearly explained how the error metrics are computed in the reported results - for example, for table 2, an a particular range like 096 - is this computed averaged across predictions for all time steps 1 to 96, or just the last time step?


- The justification for adding the log transform in the loss seems lacking and not so clear and not really tied to the motivating points.
  - I.e., it's posed that existing approaches (using MSE loss between Nth order n-step prediction and ground truth) for fine-tuning suffer from error accumulation, but it's not clear how the proposed approach addresses this, especially since they end up using the same base error in the loss in the end. They instead state the goal of including more order error terms (i.e., at each time step as opposed to just the last), claiming without any substantiation that this can improve performance - but essentially state it's more efficient to use their approach to approximate this.  Meanwhile, prior work has used errors for each prediction step as well (as this is the typical approach for RNNs or CNNs) - which should already be doing what they are aiming to do (albeit arguably less efficiently) - and this is not compared to in the experiments.
  - In the analyses, the authors claim the Nth order error can be seen as the 1st order error plus some error accumulation term, then state that its most efficient to reduce / optimize for this Nth order error by optimizing for the 1st order error plus the "error amplifier" term (and subsequently drop the 1st order error part as well) - presumably as a way to bring in the other error terms - but the equating analysis is a bit of a stretch and requires multiple assumptions.  Additionally, they also state that the 1st order error is already accounted for by the pre-training step, so fine-tuning can focus on the error amplifier, which is then roughly approximated by the mean log squared nth order error (bringing the loss back to the original objective anyway, just with a log term).  This seems like an unjustified assumption, since fine-tuning a neural net can cause the original concept learned to be forgotten, but I would suspect that since the nth order error depends on the 1st order error anyway, it would likely avoid the first order error getting worse, but it's not justified in this way here.


- Experimental procedure and results seems a bit lacking.
  - As mentioned, specifics around hyper parameters and how they are selected are missing.
  - For experiment results, it's hard to tell if the difference shown is significant, because the differences generally seem very small, no error bars are reported, and apparently a single sample is used to get the test error scores.  Along with not knowing how methods are tuned, it's hard to say if the improvements shown aren't from over-fitting to the particular test sample.  In general it's hard not to be skeptical of the results in light of the lack of details, given the very minor change applied.
  - It would also be better to see the mean and variance of the results over multiple test sets (test windows) and model training runs - e.g., with time series cross-validation (sliding window evaluation)
   - It seems like the e2e approaches are not applied as e2e - so it is not really a fair comparison - as it's stated they are applied auto-regressively after the initial output window, which defeats the purpose.
  - Another thing that casts some doubts on the correctness of results, is that the forecasts for the 0-96 output horizon for which the e2e models are specifically trained does worse than the auto-regressive approach fine-tuning those same models for additional future horizons, according to the results (Table 2).  This doesn't seem to make much sense - since the fine-tuning is causing the models to focus on future errors which the models are not even evaluated on, and this would typically only match the targeted models performance or possibly make it worse.  Yet somehow in these results it always improves it.  However, this would trivially make sense, if the metric being evaluated in this case is the prediction only on the last timestep in that window (i.e., at time step 96) as the fine-tuning would cause the model to focus more on predicting that time step in particular more accurately (at the expense of other time steps like 95, 94, etc.) - so these results are not really showing anything when it comes to comparing to E2E.  Since for the E2E case we also care about predicting accurately all the time steps before the last one being predicted in that prediction window - and these models were not given the opportunity to be trained focusing on one particular time step - so it's also not a fair comparison.

**Questions:**

- Please see the detailed comments in the weaknesses section above.

---

> ### Author Response · Authors · 2023-11-16
>
> Thank you for your invaluable input. However, there seems to be a slight **misunderstanding** regarding the methodology employed in our paper.
>
> The reviewer seems to be under the impression that we have used **only the highest order error** term for model optimization. This **is not the case**. In actuality, we have utilized all error terms during the training process. The term "fine-tuning" in our context refers to **optimizing the entire trajectory**, rather than a single step.
>
> As such, the concern that `fine-tuning a neural network may lead to the original concept being forgotten`, or that` it would cause the model to focus excessively on predicting a particular time step`, is not applicable in this instance. Our model is designed to perceive and correct for 1st order error consistently, ensuring that it does not face an extreme 'forgetting' challenge.
>
> Additionally, we would like to clarify that **our experiments were conducted with utmost care to ensure fairness.** The correct prediction sequence was always used for error computation and comparison. We strive for transparency in our research and **stand by the fairness** of our experimental design. We believe that these clarifications address your concerns, and we appreciate your engagement and insightful feedback.
>
> ----
> ### One by one response:
>
> Response to **Weaknesses.1** : We would like to clarify a misunderstanding: The loss function used in our study is the **Mean Log Squared Error** (MLSE)$\sum ln(x_i^2)$, **not the Mean Squared Logarithmic Error** (MSLE) $\sum(ln(1+x_i)-ln(1+\bar{x}))^2$. These are fundamentally different concepts with different applications and interpretations. Our choice of MLSE was deliberate and based on the specific objectives and demands of our research. To the best of our knowledge, there has been no explicit discussion of the influence of MLSE in similar contexts. Therefore, we believe our exploration of this area offers a unique contribution to the field.
>
> Response to **Weaknesses.2** : We appreciate your interest in the detailed experimental setup. However, we believe there might be some confusion regarding the error calculation process we deployed. In  table 2, our methodology involves calculating the error as an average across all time step predictions, such as 0 -> 96, 0->196, etc. This approach aligns with the paradigm used in the previous work like TimesNet, **ensuring a fair basis for comparison** between our ATT results and E2E results.  A more clean clarify below:
> - Train
>   - the E2E model train on setting that use `seq[0:96]` to predicted `seq[96:96+720]`
>   - the ATT model trained on setting that used `seq[0:96]` to predict `seq[96:192]` and autoregressive for seq[192:192+96].
> - Testing:
>   - the E2E use `seq[0:96]` to predicted `seq[96:96+720]` in one step
>   - the ATT use `seq[0:96]` to predicted `seq[96:96+720]` in 720//96 + 1 times
>   The error is compute via take average for each timestamp prediction for `label[96:96+720]`
>
> Response to **Weaknesses.3.a**: The key challenge of long-term autoregressive optimization lies in the substantial resources required for storing optimizer-intermediates. For a more in-depth technical understanding, we refer you to studies such as Pangu, Fengwu, and GraphCast. To illustrate this challenge, consider the process of iterating the model $f$ multiple ($N$) times to constrain the long-term autoregressive error. When using an autodifferential engine to compute the gradient, it necessitates the storage of $N$ times the optimizer state. In a scenario where we use the ViT architecture (usually at least 400M parameters) and consider one image input as a (70, 1440, 720) image (a normal case for Weather data), it would require approximately `40G`*`N` of GPU memory per update even under half precision. This presents a significant challenge when training a skillful long-term prediction model. Our research offers a unique solution to this challenge. We found that by merely changing the form of the loss from Mean Squared Error (MSE) to Mean Log Squared Error (MLSE), we can enhance performance using a smaller $N$. This approach significantly reduces the resources required for training while maintaining, or even improving, performance.
>
> Response to **Weaknesses.3.b**: Kindly notice the fine-tuning procedure optimizes both the $\mathcal{E}^I$ and $\mathcal{E}^{II}$ error terms. This is in contrast to the pretraining phase, which only optimizes $\mathcal{E}^I$. Therefore, the fine-tuning phase does not cause the model to forget the original concept learned but instead enriches it with additional information pertaining to $\mathcal{E}^{II}$.
>
> Response to **Weaknesses.4**:  To address your query, we have provided extensive information on the experiment setting, including the hyperparameters and the 'mean and variance' in the appendix. Specifically, the hyperparameters are outlined on pages 13 and 14, and a snapshot view of the 'mean and variance' can be found on pages 9, 10, and 11.

---

### Meta-Review · Area_Chair_pqEK · 2023-12-07

**Metareview:**

The paper advocates for the use of MLSE loss for forecasting, and benchmarks the MLSE loss with a autoregressive pre-trained/fine-tuned strategy on commonly used forecasting architectures. The loss function, while not really novel,  is a very simple and practical modification, and showcases gains over using standard MSE loss. Reviewers however expressed concerns (that AC concurs with) about the lack of novelty coupled with the lack of a comprehensive empirical evaluation over more datasets and over more baseline loss functions beyond MSE.  Especially considering the new MAE experiments, there needs to be a more expansive evaluation to ensure the results are not overfitting to the small set of test benchmarks.

**Justification For Why Not Higher Score:**

This is a primarily empirical paper with very little novelty (the proposed loss function has been used before, although not commonly, in the forecasting community). Considering that, the empirical evaluation is very weak. The authors do not compare with sufficiently strong loss function baselines, and on a sufficient span of datasets.

**Justification For Why Not Lower Score:**

N/A

---

### Decision · Program_Chairs · 2024-01-16

Reject